# Lipidomic and biophysical homeostasis of mammalian membranes counteracts dietary lipid perturbations to maintain cellular fitness

Kandice R. Levental [1✉], Eric Malmberg[1], Jessica L. Symons[1], Yang-Yi Fan[2], Robert S. Chapkin[2], Robert Ernst [3] & Ilya Levental [1✉]

Proper membrane physiology requires maintenance of biophysical properties, which must be buffered from external perturbations. While homeostatic adaptation of membrane fluidity to temperature variation is a ubiquitous feature of ectothermic organisms, such responsive membrane adaptation to external inputs has not been directly observed in mammals. Here, we report that challenging mammalian membranes by dietary lipids leads to robust lipidomic remodeling to preserve membrane physical properties. Specifically, exogenous poly-unsaturated fatty acids are rapidly incorporated into membrane lipids, inducing a reduction in membrane packing. These effects are rapidly compensated both in culture and in vivo by lipidome-wide remodeling, most notably upregulation of saturated lipids and cholesterol, resulting in recovery of membrane packing and permeability. Abrogation of this response results in cytotoxicity when membrane homeostasis is challenged by dietary lipids. These results reveal an essential mammalian mechanism for membrane homeostasis wherein lipidome remodeling in response to dietary lipid inputs preserves functional membrane phenotypes.

[1] Department of Integrative Biology & Pharmacology, University of Texas Health Science Center at Houston, Houston, TX, USA. [2] Program in Integrative Nutrition & Complex Diseases and Department of Nutrition, Texas A&M University, College Station, TX, USA. [3] Department of Medical Biochemistry & Molecular Biology, Medical Faculty, Saarland University, Homburg, Germany. ✉email: kandice.r.levental@uth.tmc.edu; ilya.levental@uth.tmc.edu

Lipidic membranes are the essential barriers between life and the abiotic world and also mediate most intracellular compartmentalization in eukaryotic cells. However, the functions of membranes are not limited to passive barriers. Approximately one third of all proteins are membrane embedded[1], and many more are membrane-associated through post-translational modifications, lipid binding, and protein–protein interactions. Thus, a major fraction of cellular bioactivity occurs at membrane interfaces. The physicochemical properties of this lipid matrix are key contributors to membrane physiology. Canonical examples are membrane viscosity, which determines protein diffusivity and protein–protein interaction frequency, and membrane permeability, which governs the diffusion of solutes into and out of the cytosol. Numerous other membrane physical parameters can determine protein behavior, including but not limited to fluidity, permeability, curvature, tension, packing, bilayer thickness, and lateral compartmentalization[2–9].

Because of their central role in protein function, effective maintenance of membrane properties is essential for cell survival in complex and variable environments. In ectothermic (i.e. non-thermoregulating) organisms, a pervasive challenge to membrane homeostasis comes from temperature variations. Low temperature reduces the motion of lipid acyl chains, causing membranes to laterally contract, stiffen, and become more viscous[10]. Organisms across the tree of life, from prokaryotes to ectothermic animals, respond to such perturbations by tuning membrane lipid composition, down-regulating tightly packing lipids (e.g. containing saturated acyl chains) and up-regulating more loosely packed ones containing unsaturations or methylations in their lipid tails[10–12]. This response was termed "homeoviscous adaptation", as these lipid changes result in remarkable constancy in membrane fluidity in spite of variable growth conditions[10,11]. It is worth noting that while fluidity is maintained at a specific set point, it should not be assumed that it is either the control variable or the physical property being sensed; rather membrane fluidity may correlate with other membrane/lipid control parameters. For example, membrane homeostasis in *Bacillus subtilis* is mediated by the DesK sensor that is believed to sense membrane thickness[13]. In the yeast endoplasmic reticulum (ER), recently discovered sensors are sensitive specifically to lipid packing[5] and bilayer compressibility[14].

Excepting a few specialized instances (e.g. hibernation), mammals and other warm-blooded animals are not subject to large-scale variations in body temperature; thus, there has been relatively little investigation of homeostatic membrane responsiveness in such organisms. However, it is a well-established but under-appreciated fact that mammalian membrane homeostasis is extensively challenged by dietary inputs. Dietary lipids have major impacts on membrane compositions in vivo[15,16] and these perturbations must presumably be buffered to maintain cellular functionality. Mammalian lipidomes are much more complex[17–19] than either bacteria[20] or yeast[21,22], suggesting more potential control nodes required to balance the various conflicting demands of mammalian membrane physiology.

The possibility of a homeoviscous response in mammalian cells was suggested in the 1970s by studies of a spontaneously arising mutant of Chinese Hamster Ovary (CHO) cells defective in cholesterol regulation[23]. This mutant accumulates cholesterol compared to wild-type CHO cells, but maintains normal membrane fluidity, possibly through modulation of phospholipid profiles. However, the molecular etiology of the defects in this mutant remains unknown, and it was not reported where the cholesterol in these cells was accumulating (possibly storage organelles or lysosomes). Further, limitations of then-available technologies prevented direct demonstration of lipidomic responses to cholesterol modulation. Thus, the relevance of those insights to physiologically relevant perturbations of metabolically normal mammalian cells remains unclear. More recently, homeoviscous adaptation in mammals has been inferred from data-driven modeling approaches, which used the physical properties (melting temperature, intrinsic curvature) of pure lipids to extrapolate those of complex, biological membranes[24]. However, the inherent non-additivity[25] and non-ideality[26] of lipid mixtures suggests that extrapolation of physical parameters of complex membranes from pure lipids may not be a reliable approach. Finally, lipid composition and membrane properties have been implicated in the heat shock response, though usually with a specific focus on signaling of the proteostasis network[27].

Here, we directly evaluate the hypothesis that mammalian membranes homeostatically adapt to dietary inputs by characterizing the lipidomic and biophysical responses to fatty acid supplementation in cultured mammalian cells and in vivo. Using shotgun mass spectrometry, we show that polyunsaturated fatty acids (PUFAs) are robustly incorporated into membrane phospholipids, introducing significant biophysical perturbations. This perturbation is counterbalanced by rapid lipidomic remodeling, most notably in the upregulation of saturated lipids and cholesterol. This remodeling normalizes membrane permeability and lipid packing, as evaluated by spectral imaging of the solvatochromic dye C-Laurdan. These responses are mediated in part by transcriptional sterol-regulatory machinery involving the Sterol Regulatory Element-Binding Protein 2 (SREBP2), whose genetic or pharmacological inhibition abrogated lipidomic and biophysical homeostasis. Finally, we show that the homeostatic membrane response is essential for cellular fitness, as uncompensated lipidomic perturbations lead to increased membrane permeability and non-apoptotic cell death when membrane homeostasis is challenged by exogenous fatty acids.

## Results

**Robust incorporation of PUFAs into membrane lipids**. Recent observations revealed that supplementation of cultured mammalian mast cells (rat basophilic leukemia cells (RBL)) with docosahexaenoic acid (DHA) leads to robust incorporation of this dietary PUFA into membrane lipids[19]. We observed similar effects in isolated human mesenchymal stem cells[28], cultured CHO cells, and rat primary hippocampal neurons (Supplementary Fig. 1), confirming that uptake and incorporation of exogenous DHA into membrane lipids is not cell-type specific. Supplementation designed to recapitulate DHA-enriched diets in mammals increased the fraction of DHA-containing glycerophospholipids (GPLs) by nearly 15-fold (from <1 to ~15 mol%) (Fig. 1a). Similarly, supplementation with the more common PUFA ω-6 arachidonic acid (AA) increased the fraction of AA-containing lipids by >3-fold (Fig. 1b). Surprisingly, supplementation with monounsaturated (oleic; OA) or saturated (palmitic; PA) fatty acids produced only minimal lipidomic changes (Supplementary Fig. 2). We expect that this disparity is associated with the availability of these fatty acids in cell culture media:[29] cultured cells have access to sufficient OA and PA such that supplementation at concentrations used here has no effect, whereas PUFA levels are limited such that supplementation with the physiologically appropriate concentrations used here leads to robust uptake and incorporation.

To confirm that the membrane incorporation of supplemented PUFAs in cultured cells appropriately recapitulates in vivo conditions, we analyzed membrane lipidomes in mouse cardiac tissue after 2 weeks of ad libitum-feeding on a semi-purified diet containing either fish oil (FO) or corn oil (CO), as previously described[30]. The FO diet is highly enriched in long-chain PUFAs, with ω-3 DHA and eicosapentaenoic acid (EPA; 20:5) comprising

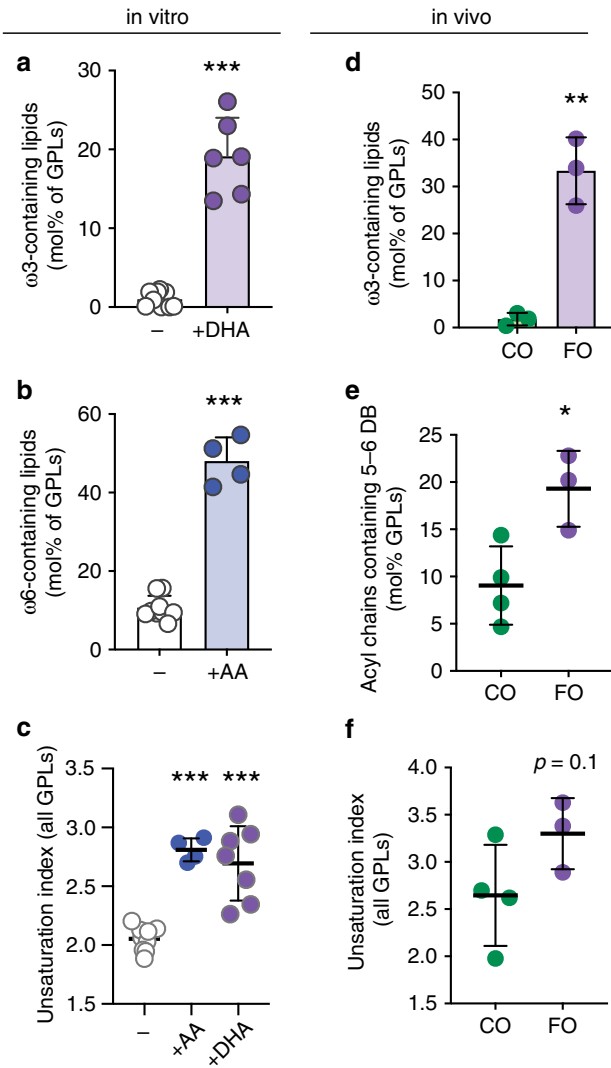

**Fig 1 Supplemented PUFAs are robustly incorporated into membrane phospholipids in vitro and in vivo.** Supplementation of culture media with PUFAs for 3 days (20 μM) leads to a dramatic increase in (**a**) ω−3 PUFA-containing membrane lipids in RBL cells supplemented with DHA and (**b**) ω−6 PUFA-containing membrane lipids supplemented with AA. (**c**) Both treatments result in significantly increased overall unsaturation of membrane lipids. The unsaturation index reflects a concentration-weighted average lipid unsaturation. (**d**) Mice fed a diet rich in fish oil (FO) have significantly more lipids containing ω−3 PUFAs compared to corn oil (CO) fed mice. (**e**) Incorporation of dietary ω−3 PUFAs increases cardiac tissue lipids containing very highly unsaturated (5 or 6 double bonds) acyl chains, resulting in an (**f**) increase in the overall unsaturation of membrane glycerophospholipids. Individual experiments (**a**–**c**) or animals (**d**–**f**) are shown. Bars represent mean ± SD. **$p < 0.01$, ***$p < 0.001$ for unpaired $t$ test compared to untreated. Treatment with saturated (PA) or monounsaturated (OA) fatty acids in these conditions had no effect on the lipidome (see Fig. S2). Source data are provided as a Source Data file.

~20% of all fatty acids. In contrast, CO is poor in long-chain PUFAs (1.3% FAs with 3+ unsaturations), with ω−6 linoleic acid (18:2) being the major PUFA source (>50% of dietary FAs) (see Supplementary Table 1 for full dietary lipid profile). Consistent with previous observations[30–32] and the in vitro measurements here, dietary FO supplementation produced a robust incorporation of ω−3 PUFAs into cardiac membrane lipids, with ~18-fold more ω−3 PUFA-containing membrane lipids in the FO-fed

compared to CO-fed tissues (Fig. 1d). As a result, FO-fed mice had a significantly higher proportion of lipids containing long, highly unsaturated acyl chains (5 or 6 double bonds; Fig. 1e), which resulted in an overall increase in unsaturation of FO-fed tissues (Fig. 1f).

**Lipidome remodeling associated with PUFA incorporation.** Despite the copious incorporation of exogenous PUFAs into membrane lipids and the resulting increase in overall membrane unsaturation (Fig. 1), cells in vitro did not show any obvious toxicity or differences in proliferation at these levels of supplementation (Supplementary Fig. 3). This observation was somewhat surprising in light of the central role of lipid unsaturation in membrane physical properties[33,34] and the putatively critical role of those properties in various cellular processes[3,4,35]. Thus, we hypothesized that mammalian cells may compensate for perturbations from exogenous FAs by remodeling their lipidomes. Indeed, both AA and DHA supplementation reduced the abundance of other polyunsaturated (i.e. di- and tri-unsaturated) lipid species (Fig. 2a). This effect could potentially be explained by replacement of these PUFA-containing lipids by AA/DHA-containing ones. Much more surprising was the highly significant, approximately 2-fold increase in fully saturated lipids resulting from PUFA supplementation (Fig. 2a), due to the significantly increased abundance of phospholipid-incorporated saturated fatty acids (Fig. 2a, inset). These changes led to significantly reduced overall unsaturation in phospholipids not containing the supplemented FAs (Fig. 2b, c, see figure legend for details). These effects were, at least in part, transcriptionally mediated, as evidenced by mRNA levels of the major fatty acid desaturase enzymes SCD1 and SCD2. As previously reported[36], both were substantially down-regulated by DHA treatment (Supplementary Fig. 4), consistent with more saturated and fewer polyunsaturated lipids. The headgroup profile of membrane lipids was not notably affected by any of the treatments (Supplementary Fig. 5).

This lipidomic remodeling in response to PUFA incorporation in vitro was not limited to a single cell type. The observations described so far were on a transformed leukocyte cell line (RBL); however, very similar DHA-induced lipid changes were observed in cultured CHO cells, primary rat hippocampal neurons, primary human MSCs, and MSCs that were differentiated in vitro into osteoblasts or adipocytes (Fig. 2d). Both the broad trends of the lipidomic remodeling (more saturated, less unsaturated lipids) and the magnitude of the effects were quantitatively similar between these disparate cell lineages and sources, despite drastic differences in overall lipid composition[19,28,37,38]. The similarity of the responses between freshly isolated cells, cultured primary cells, and long-term cultured cell lines derived from two different germ layers and three different organisms suggests that comprehensive lipidomic remodeling in response to exogenous PUFAs is a general phenomenon for mammalian cells.

Importantly, a similar response is also evident in vivo. The incorporation of highly polyunsaturated PUFAs from the diet into membrane lipids in mouse cardiac tissue (Fig. 1d, e) led to remarkably similar lipidomic remodeling as in cultured cells. Namely, we observed that FO-fed mice have ~2-fold more saturated lipids compared to CO-fed mice, consistent with compensatory, homeostatic lipidomic remodeling in vivo (Fig. 2e). The striking differences between FO and CO feeding are notable in light of the fact that CO diets do contain abundant PUFAs in the form of linoleic acid (18:2). These differences are potentially due to the unique effects of highly polyunsaturated lipids on membrane properties, which scale with the degree of unsaturation[38,39]. An alternative, non-exclusive possibility is that

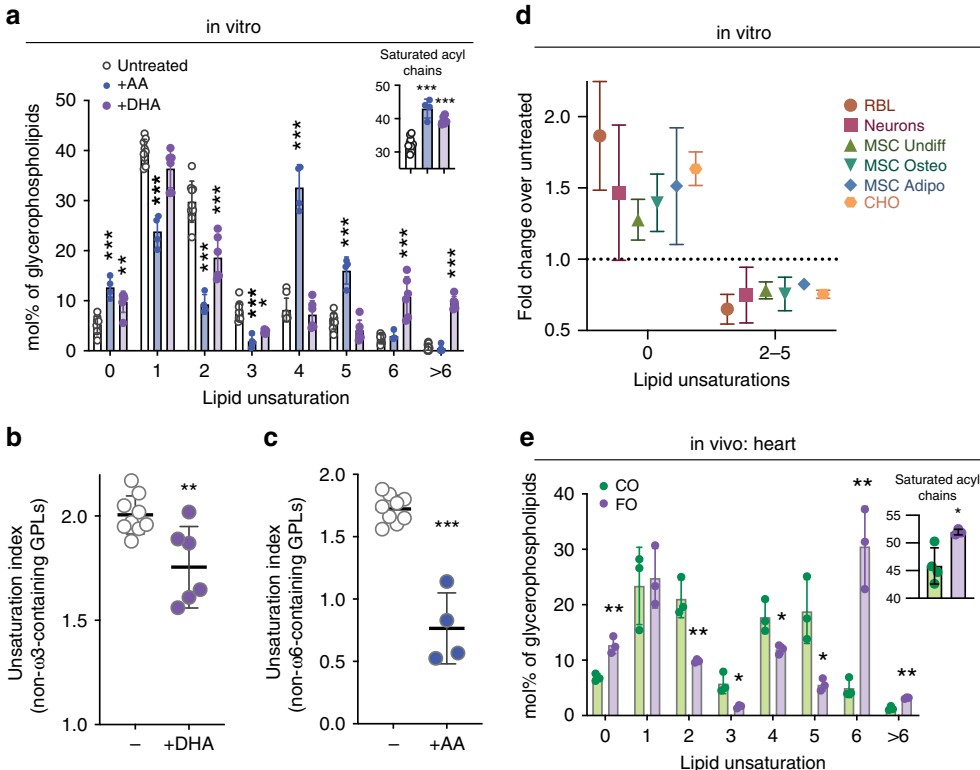

**Fig. 2 Lipidome remodeling induced by PUFA incorporation. a** Lipidome-wide remodeling of lipid unsaturation induced by DHA or AA supplementation. Both PUFAs induce significantly increased saturated lipids and decrease lipids containing di- and tri-unsaturated lipids. Inset shows the mol% of fully saturated acyl chains in GPLs. **b**, **c** The unsaturation index (concentration-weighted average lipid unsaturation) of lipids not containing the supplemented FAs and their derivatives is significantly reduced upon PUFA supplementation. For accurate estimates of lipidomic remodeling, the supplemented FAs and their derivatives are removed from the analysis, as the rather extreme over-abundance of those species upon supplementation suppresses the visualization of compensatory effects (e.g. for DHA supplementation, lipids containing 22:6, 20:5, or 24:6 are removed and then the unsaturation index of the "remaining" lipids is calculated). **d** DHA-mediated lipidomic remodeling, indicated by increased saturated lipids and decreased polyunsaturated lipids (2–5 unsaturations), is consistent across multiple cell types, including isolated rat hippocampal neurons, cultured human MSCs, and MSCs differentiated into adipogenic or osteogenic lineages. **e** Lipid unsaturation profile in membrane lipids isolated from murine heart tissue after feeding with CO versus FO. Incorporation of ω-3 PUFAs into membrane lipids (see Fig. 1d) is associated with lipidomic remodeling, namely higher levels of saturated lipids. Inset shows the mol% of fully saturated acyl chains in GPLs. All data are average ±SD for $n \geq 3$ biological replicates. (**a**) and (**e**) are two-way ANOVA with Sidak's multiple comparison test. (**b**, **c**) and insets are student's *t* test compared to untreated. *$p < 0.05$, **$p < 0.01$, ***$p < 0.001$. Source data are provided as a Source Data file.

highly unsaturated FAs are preferentially taken up into membrane lipids. These details will be resolved by future studies with various FA feeding protocols.

**Rapid lipidomic remodeling after PUFA incorporation**. The remodeling associated with PUFA incorporation into membrane lipids suggested the induction of a homeostatic membrane response, wherein saturated lipids are upregulated to compensate for the fluidizing effect of PUFA-containing species. To support this inference, we analyzed the temporal profiles of PUFA incorporation and associated lipidomic changes in vitro. DHA was quickly incorporated into membrane lipids, with significant increases in DHA-containing lipids observed within 1 h of supplementation and half-time of incorporation of ~4 h (Fig. 3a, b). The recovery time course was significantly slower, as wash-out of DHA was followed by a return to baseline with a half-time of ~25 h (Fig. 3c). This is the approximate doubling time of RBL cells in culture, suggesting that the wash-out effect likely results from dilution by new lipid synthesis rather than directed removal of DHA from membrane lipids. Remarkably, the associated lipidome remodeling proceeded with similar temporal profiles to both DHA supplementation and wash-out (Fig. 3d–i). The

increase in saturated lipids (Fig. 3e) and the decrease in di-unsaturated lipids (Fig. 3h) were observed shortly after DHA incorporation, with similar time courses for the wash-out (Fig. 3f–i). These observations reveal unexpectedly abrupt lipidomic perturbations which induce rapid compensatory responses.

The broad compensatory lipidomic remodeling in response to PUFA supplementation in mammalian cells evokes comparisons to classical observations of homeoviscous adaptation in ectothermic organisms[10–12,40,41]. There, perturbations of membrane physical properties produced by changes in ambient temperature induce lipid changes to re-normalize membrane physical properties. Our findings suggest that a similar response to fluidizing stimuli occurs in mammalian cells, where it has been co-opted to cope with perturbations from exogenous lipids (e.g. from the diet).

**Cholesterol upregulation by DHA via SREBP2.** Acyl chain remodeling in response to PUFA supplementation was incomplete, as the unsaturation index of PUFA-supplemented cells remained higher than those from untreated controls (Fig. 1c–f). If the purpose of this lipidome response is to normalize membrane properties, the acyl chain remodeling appears insufficient on its

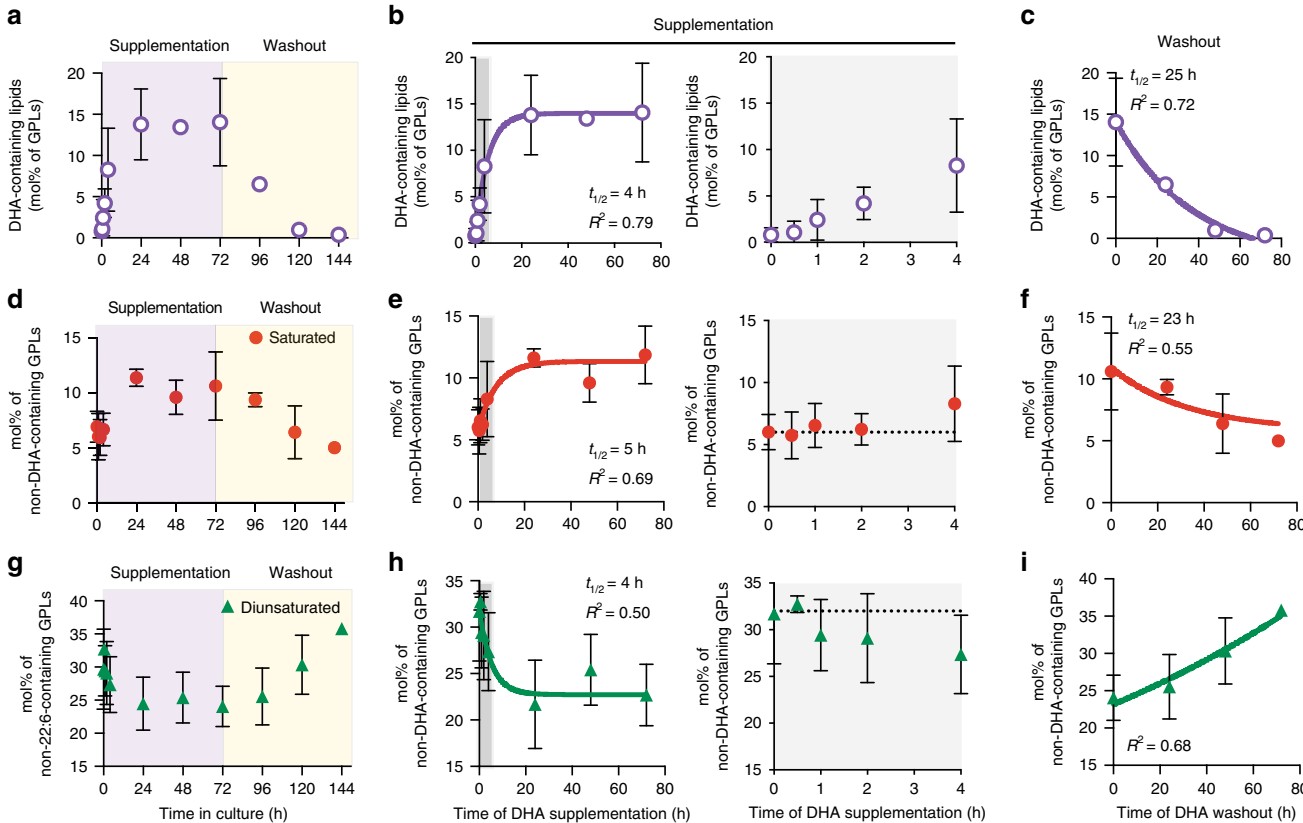

**Fig. 3 Time course of DHA incorporation and lipidome remodeling. a–c** Time course of DHA incorporation into GPLs following supplementation, then wash-out. Time course of nearly concomitant (**d–f**) saturated lipid and (**g–i**) di-unsaturated lipid changes induced by DHA supplementation. The gray areas in (**b**), (**e**), and (**h**) are shown in extended charts to the right of each panel to illustrate the timing of the remodeling. All data shown are average ±SD for $n \geq 3$ biological replicates. All fits shown are first-order kinetic models. Source data are provided as a Source Data file.

own (lipid headgroup remodeling was minimal; Fig. S5C). In contrast to most prokaryotes wherein membrane homeostasis is mediated by phospholipids[11], membrane properties in eukaryotes are regulated to a very significant extent by sterols[42]. Thus, we hypothesized that the adaptive response to PUFA perturbation in mammalian cells was also mediated by cholesterol. Indeed, DHA supplementation led to significantly increased membrane cholesterol in both FO-fed murine heart tissue (Fig. 4a) and cultured cells (Fig. 4b). In cells, the increase in cholesterol abundance was again quite rapid, with effects observed within 1 h of DHA introduction and reaching a peak after ~4 h (Fig. 4c), essentially concomitant with the acyl chain remodeling (Fig. 3), suggesting that cholesterol upregulation is an early and potent response to membrane perturbation.

The machinery for cholesterol production in metazoans is regulated by proteolytic processing of transcription factors of the SREBP family[43]. Specifically, signals to upregulate cholesterol levels lead to proteolysis of a membrane-bound SREBP2 precursor to release a "mature" cleaved fragment, which translocates to the nucleus to induce transcription of various target genes, including those for cholesterol synthesis and uptake[43]. Having observed a robust and rapid increase in cholesterol levels resulting from DHA supplementation, we evaluated whether SREBP2 processing was associated with this response. Indeed, DHA feeding increased the production of the "mature" transcription factor form of SREBP2, with minimal effect on the precursor form (Fig. 4d). In contrast, the proteolytic processing of SREBP1, the SREBP family member primarily implicated in unsaturated fatty acid metabolism[43], was not affected by DHA under our conditions (Fig. 4e). The relationship

between these observations and previous reports of PUFA-regulation of SREBPs are expanded upon in the Discussion.

Our findings suggested that processing of SREBP2 is involved in homeostatic membrane lipid remodeling in response to perturbation by PUFA incorporation. To test this inference, we measured the response to PUFA supplementation in cells whose SREBP processing was inhibited either genetically or pharmacologically. SRD-12B cells are clonal variants of CHO cells wherein a genetic defect in the Site-1 Protease (S1P; cleaves SREBPs to produce transcriptionally active forms) prevents SREBP processing/activation[44]. In these cells, no notable upregulation of cholesterol (Fig. 4f) or saturated lipids (Supplementary Fig. 7) was observed upon DHA supplementation. Similarly, chemical inhibition of SREBP activation by the pentacyclic triterpene betulin[45] abrogated DHA-induced upregulation of cholesterol (Fig. 4b). It is important to point out that S1P also processes other important signaling proteins like ATF6, responses that would be abrogated in SRD-12B cells (though likely not by betulin treatment); however, we did not observe DHA-dependent cleavage of ATF6 (Supplementary Fig. 8a), nor a significant upregulation of the ATF6 target gene Grp78. Finally, we did not observe induction of HSP90B or processing of XBP1, suggesting that ER stress is not significantly induced by DHA feeding (Supplementary Fig. 8b). While these results support a role for SREBP2 processing in PUFA-induced lipidome remodeling, they explicitly do not rule out other contributors to the homeostatic response, nor do they identify SREBP2 as the sensor for membrane fluidity. Rather, they establish SREBP2 as an important node of the homeostatic response, which can be manipulated to evaluate the cellular consequences of disruptions in membrane homeostasis.

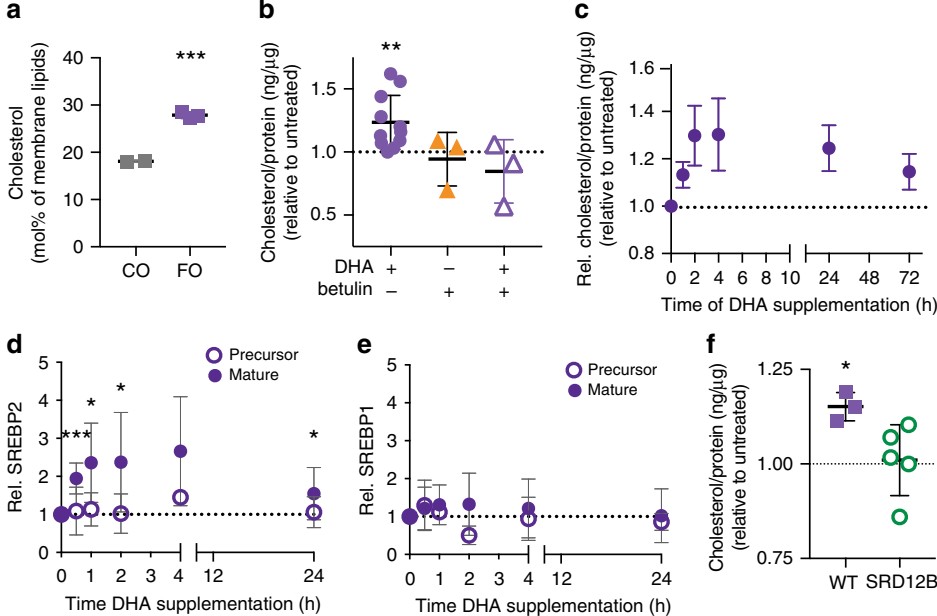

**Fig 4 Cholesterol upregulation by DHA supplementation. a** Cholesterol levels are higher in membranes isolated from murine heart tissue after feeding with FO compared to CO. **b** Membrane cholesterol is significantly increased by DHA supplementation of cultured RBLs. Effect is abrogated by simultaneous treatment with 200 nM betulin. **c** Time course suggests cholesterol increase is a rapid response to DHA-mediated membrane perturbation. **d** The "mature", transcription-competent form of SREBP2 is rapidly produced in response to DHA supplementation; $n > 6$. **e** SREBP1 proteolytic processing is not significantly affected by DHA supplementation; $n = 4$. **f** Cholesterol is significantly increased in WT CHO cells treated with DHA; this effect is abrogated in cells with a defect in SREBP activation (SRD12B; S1P-negative). All data shown are average ±SD for $n \geq 3$ biological replicates; ***$p < 0.001$ in (**a**) is one-sample $t$ test between groups; *$p < 0.05$, **$p < 0.01$, ***$p < 0.001$ in (**b**), (**d**), and (**f**) are one-sample $t$ tests compared to untreated. Full representative Western blots shown in Supplementary Fig. 6. Source data are provided as a Source Data file.

One possible explanation for the unexpectedly fast cholesterol response to early stages of DHA accumulation (e.g. 1 h after supplementation) is that DHA-containing lipids are highly fluidizing, even at relatively minor relative abundances[19]. Further, these are likely produced first in the ER and may accumulate at the site of synthesis to much higher levels than in total cell membranes. This local perturbation may induce a rapid and potent response by the cholesterol biosynthetic machinery, which has non-transcriptional as well as transcriptional regulation[46]. It is possible that the early phase of this response is mediated by non-transcriptional regulation of already-available enzymes, whereas the latter phases are dependent on transcriptional upregulation.

**Physical homeostasis following lipidomic perturbation.** The above data reveal rapid and comprehensive lipidomic remodeling wherein lipids that decrease membrane fluidity and increase membrane packing (e.g. saturated lipids and cholesterol) are upregulated in response to introduction of PUFA-containing lipids that increase membrane fluidity. These changes are consistent with lipidome remodeling for homeostatic maintenance of membrane physical properties. To directly evaluate this inference, we measured membrane packing in live cells using a fluorescence assay that relies on a solvatochromic dye (C-Laurdan) whose spectral characteristics are dependent on membrane properties[47]. Specifically, the emission spectrum of C-Laurdan is red-shifted in loosely packed membranes (due to the enhanced polarity of the fluorophore nano-environment), and the relative extent of this spectral shift can be quantified by ratiometric spectroscopy or imaging[48]. The resulting dimensionless parameter called Generalized Polarization (GP) is a widely used proxy for membrane packing and fluidity[49–51], with higher values reporting more tightly packed membranes.

The GP of homogenized cell membranes was not significantly affected by 24 h supplementation with DHA, nor by betulin alone. In contrast, the combination of DHA + betulin significantly reduced lipid packing (Fig. 5a). Similarly, while wild-type CHO cells did not show a significant effect of DHA on membrane fluidity after 24 h treatment, DHA supplementation of SRD-12B cells significantly reduced GP (Fig. 5b). These measurements of lipid packing in homogenized whole cell membranes suggested that inhibition of lipidomic remodeling prevents the homeostatic maintenance of membrane physical properties.

To gain temporal and spatial insight into DHA-mediated lipidomic remodeling, we used confocal spectral imaging[52] to image C-Laurdan GP in cells. Figure 5c shows two-dimensional GP maps of DHA and/or betulin-treated RBL cells with relatively fluid (low GP) internal membranes and relatively packed plasma membranes (high GP) characteristic of mammalian cells[48]. The clearly distinct GP regions enabled us to separately quantify the effects of DHA on plasma versus internal membranes. Figure 5d, e shows the GP histograms from the regions within the dotted lines in Fig. 5c, representing internal membranes. The packing of these internal membranes was significantly decreased shortly after introduction of exogenous DHA (Fig. 5f), suggesting reduced packing consistent with the expected fluidizing effect of PUFA-containing lipids[19,53]. This response eventually reversed, with GP fully normalizing to baseline by 24 h (Fig. 5f, purple). Thus, DHA incorporation combined with the associated remodeling produced no net change in overall membrane properties. GP at the plasma membrane (PM) was also not significantly affected by DHA treatment (Fig. 5g), and all lipidomic remodeling effects described above for whole cell membranes were also observed in isolated PMs[19]. Namely, supplemented PUFAs accumulate in PM lipids, which are then compensated for by increases in saturated lipids and cholesterol (Supplementary Fig. 9). One possible explanation for why PMs showed smaller biophysical changes

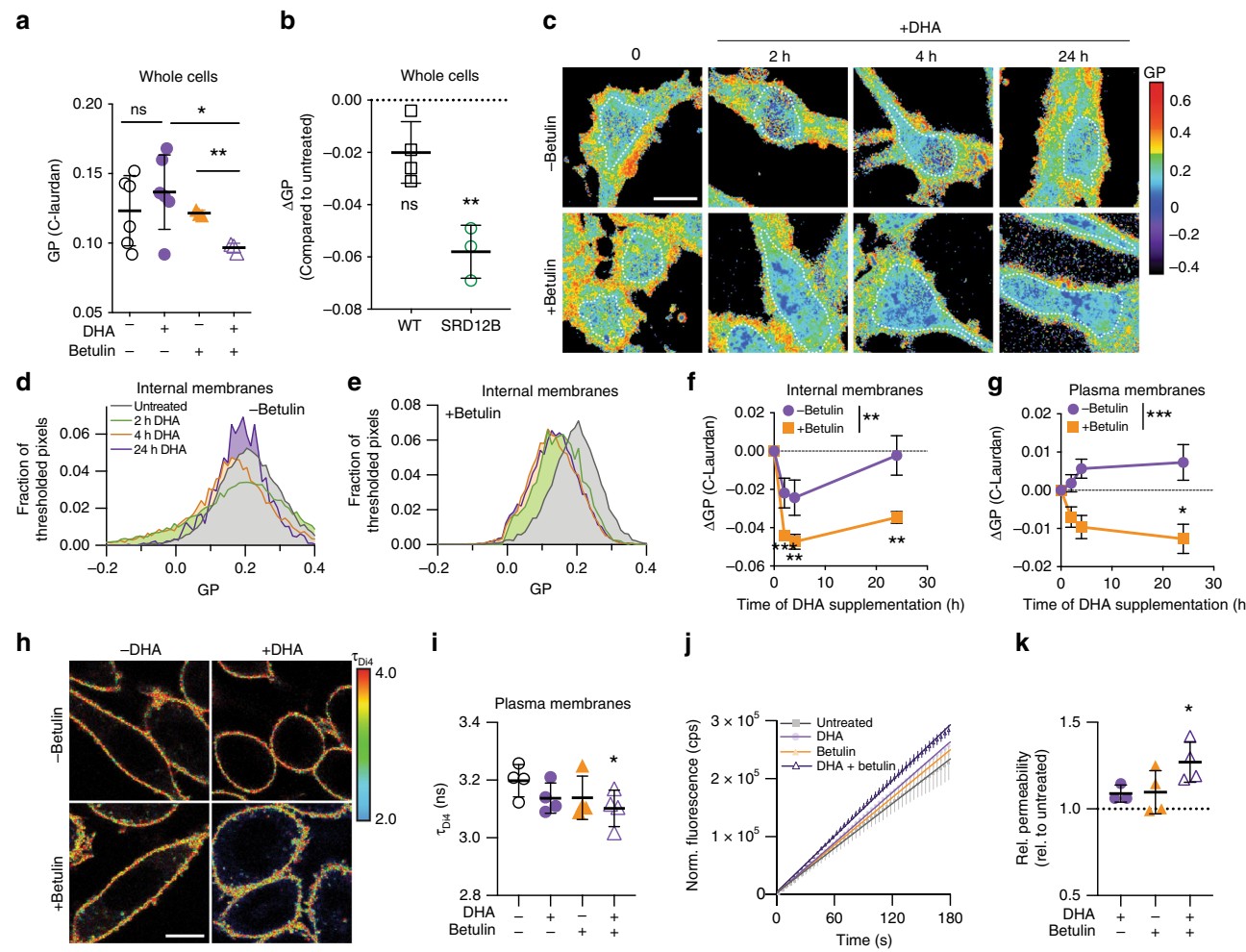

**Fig 5 Physical membrane homeostasis is disrupted by inhibition of lipidomic response. a** Spectroscopy of C-Laurdan-labeled homogenized whole cell membranes shows that neither DHA nor betulin alone affected membrane packing (GP) after 24 h, whereas DHA + betulin significantly reduced membrane packing. **b** Similarly, DHA supplementation had no effect on WT CHO cells after 24 h, whereas SRD-12B cells (defective in SREBP processing) had significantly reduced membrane packing. **c** GP maps of RBL cells treated with DHA and/or betulin demonstrate temporal changes in lipid packing (GP). The white dotted lines denote the areas from which the GP of the internal membranes was calculated. **d, e** GP histograms of internal membranes from images in **c**. Leftward shift in histograms denotes less tightly packed, more fluid membranes. **f, g** ΔGP represents GP of treated cells normalized to untreated within each experiment. Internal membranes are initially fluidized by excess DHA-containing lipids (purple), then recover to baseline by 24 h. In contrast, internal membrane packing does not recover in betulin-treated cells (orange). **g** DHA supplementation had no significant effect on packing at the plasma membrane (PM); however, when betulin is added to inhibit lipidomic remodeling, the PM also becomes significantly more fluid upon DHA supplementation. The effect of betulin treatment in PMs was significant ($p < 0.001$) by 2-way ANOVA. **h** Fluorescence lifetime imaging of Di4 to probe membrane packing in the PM. Warmer colors represent higher lifetimes (e.g. more tightly packed membranes). **i** Only the combined DHA + betulin treatment significantly reduced Di4 lifetime. **j** Exemplary plots of accumulation of fluorescein fluorescence during incubation with FDA. **k** PM permeability to FDA increases by treatment with DHA + betulin. Average ±SD for $n \geq 3$ biological replicates. $*p < 0.05$, $**p < 0.01$, $***p < 0.001$. (**a**) is a student's t test between groups. (**b**), (**d**), (**e**), and (**k**) are one-sample t tests compared to untreated. (**f**, **g**) is a two-way ANOVA to compare effect of treatment; each time point for betulin-treated cells was significantly different from zero ($p < 0.01$). (**i**) shows paired t tests compared to untreated. In (**j**), linear regressions are shown. Scale bars are 10 μm. Source data are provided as a Source Data file.

than internal membranes is that exogenous PUFAs are preferentially incorporated into lipids at the ER, which is site of most lipid synthesis. If lipidome remodeling occurs faster than the trafficking of newly synthesized lipids to the PM, then biophysical disruptions to the PM may be minimal.

The re-normalization of membrane packing was markedly abrogated when lipidomic remodeling was inhibited by targeting SREBP processing. Namely, betulin treatment suppressed the recovery of internal membrane packing following DHA treatment, leading to significantly reduced GP in DHA + betulin-treated RBL cells (Fig. 5c–f). The persistently increased fluidity

resulting from combined betulin + DHA treatment was observed not only in internal membranes, but also in PMs (Fig. 5g).

To confirm the biophysical perturbation of the PM, we employed a recently characterized method that relies on a fluorescent reporter of membrane packing (Di-4-ANEPPDHQ, Di4) that integrates solely into the outer leaflet of the PM (Fig. 5h)[54]. Similar to C-Laurdan, the photophysical characteristics of Di4 are dependent on lipid packing, with fluorescence emission lifetime providing a particularly sensitive readout, with higher lifetimes indicative of more tightly packed membranes[54,55]. While neither DHA nor betulin alone significantly affected Di4 lifetime

($\tau_{Di4}$), PM packing was significantly reduced in DHA + betulin-treated RBL cells (Fig. 5h, i).

Although biophysical membrane perturbations likely have several distinct physiological effects, we considered the central role of the PM as a selective barrier. Significant computational and model membrane literature[56,57] suggests that membrane permeability is dependent on lipid composition, with more loosely packed, fluid membranes being more permeable to various polar and amphiphilic compounds. Simulations have specifically analyzed polyunsaturated phospholipids, showing DHA-containing membranes are 2–3-fold more permeable than monounsaturated counterparts[56]. Thus, we hypothesized that the permeability of cell membranes would be increased by incorporation of polyunsaturated lipids. To test this possibility, we developed a method to quantitatively evaluate membrane permeability to amphiphilic substrates in living cells. Specifically, we adapted a classical cell viability assay that relies on passive diffusion of the amphiphilic compound fluorescein diacetate (FDA) through the PM into the cytoplasm, where cellular hydrolases convert it into fluorescent fluorescein whose charge inhibits its diffusion out of the cell[58]. Because the catalytic conversion of FDA is highly efficient[58], the rate of fluorescence increase is a direct readout of FDA flux through the PM. We observed a linear increase in fluorescence over at least the first 10 mins of measurement, consistent with a constant flux ($Q$), as shown in Fig. 5j. The permeability can then be calculated from the flux using Fick's Law (see Methods). The resulting permeability coefficient for FDA was ~$2.2 \times 10^{-6}$ cm/s, in good agreement with previously reported values in plant cells[59]. Finally, we tested the effects of DHA and/or betulin on PM permeability. Consistent with our hypothesis, cells with uncompensated membrane perturbations (i.e. DHA + betulin) were indeed more susceptible to passive permeation by FDA (Fig. 5k).

**Disruption of membrane homeostasis induces cell death.** In unicellular organisms, the homeoviscous lipidomic response is believed to be necessary for maintaining membrane physical properties in a range compatible with life[10,12]. Having observed similar lipidomic and biophysical responses in mammalian membranes, we hypothesized that this adaptation was necessary for cellular fitness following membrane perturbations. Therefore, we evaluated the cytotoxic effects of DHA under pharmacological or genetic inhibition of homeostatic responses. At low doses (up to 20 µM DHA and 500 nM betulin), neither DHA nor betulin alone were significantly cytotoxic; however, DHA supplementation in the presence of betulin led to ~50% decrease in cell number after 3 days of culture (Fig. 6a; full dose-response in Supp Fig. S10A). We then tested whether this cytostatic effect was due to reduced proliferation or increased cell death. We found that the populations of cells at various parts of cell cycle progression were unaffected by any of the treatments (Supplementary Fig. 11), suggesting that reduced proliferation rates were not the cause of lower cell numbers. We also did not observe an accumulation of sub-G0 cells, which are reflective of DNA fragmentation often associated with apoptosis. However, the combination of DHA and betulin (but neither treatment alone) significantly increased the proportion of dead (i.e. trypan blue-positive; TB+) cells (Fig. 6b). These observations were confirmed in SRD-12B cells: consistent with the abrogated homeostatic response, DHA had a significant cytotoxic effect on SRD-12B cells (Fig. 6c, d). Thus, the reduced cellular viability resulting from uncompensated membrane perturbations is due to cytotoxicity.

Consistent with increased cell death, we observed a significantly increased fraction of propidium iodide positive (PI+) cells after 24 h of combined DHA + betulin treatment (Fig. 6e and i).

Despite the lack of sub-G0 cells, we speculated that apoptosis may be involved in inducing cell death, perhaps resulting from imbalanced signaling at the PM or membrane stress induced by lipidomic perturbations. One hallmark of apoptosis is the externalization of PS (marked by Annexin V) without loss of membrane integrity (marked by PI), allowing apoptotic cells to be identified as AnxV+ /PI− by flow cytometry. After 24 h of treatment with DHA and/or betulin, we observed no significant increase in AnxV+ /PI− cells (Fig. 6f–i), suggesting that the increased cell death was not due to apoptosis. Consistently, a fluorescent marker of active caspases (CellEvent Caspase-3/7) reported no induction of apoptosis in cells treated with DHA + betulin (Supplementary Fig. 12a). Finally, the involvement of apoptosis in a cytotoxic process can be inferred by using potent caspase inhibitors (e.g. Z-VAD-FMK), which should attenuate apoptosis-induced cytotoxicity. Treatment with this inhibitor did not inhibit cell death (Supplementary Fig. 12b), supporting a non-apoptotic cytotoxicity mechanism.

Together, these results reveal that inhibition of lipidome remodeling in mammalian cells markedly increased non-apoptotic cell death upon membrane perturbation with dietary fatty acids. Because cell physiology is essentially dependent on maintaining cytoplasmic concentrations of a plethora of small molecules (water, alcohols, polyamines) and amphiphiles (steroids, fatty acids) that can passively diffuse through the membrane, it is possible that the significant increases in membrane permeability may be responsible for the observed toxicity.

## Discussion

The incorporation of dietary fatty acids into mammalian membrane lipids has been widely observed both in vitro[19] and in vivo[15,16,31,32]. That exogenous fatty acids are so readily used for lipid synthesis is unsurprising in light of the fact that the enzyme required for de novo fatty acid production (fatty acid synthase) is minimally expressed in most adult human tissues[60]. Indeed, de novo lipogenesis is considered "negligible"[61] in adult humans, suggesting that exogenous sources of fatty acids are the major raw material for maintenance and replenishment of membrane lipids. This reliance on exogenous inputs for production of components central to cellular architecture and function would seem to present a major complication for homeostasis.

In ectothermic organisms, homeostatic membrane control has been widely reported in the form of homeoviscous adaptation (HVA)[10–12]. According to this hypothesis, the relative abundance of saturated to unsaturated membrane lipids is responsive to temperature changes in order to maintain membrane fluidity at the level required for the many processes hosted and regulated by cell membranes. Although such adaptation has been proposed in prokaryotes[11,40], single-celled eukaryotes[5], and even cold-blooded animals[10], studies on endothermic organisms have been limited[23,24], and there has yet been no direct observations of HVA in mammals or isolated mammalian cells. Our data directly confirm all major tenets of cell-autonomous homeoviscous adaptation: (1) lipidomic remodeling resulting from a perturbation of membrane physical properties (Fig. 2–4); (2) recovery of baseline physical properties at a new lipid composition (Fig. 5); and (3) necessity of this response for cellular fitness (Fig. 6). These observations suggest that mammalian cells possess the capacity for homeostatic membrane adaptation analogous to HVA, and that this response can compensate for dietary lipid perturbations.

In cold-blooded animals, the homeostatic membrane response also involves modulation of cholesterol levels[42], and we observe a similar response in mammalian cells, mediated at least partially

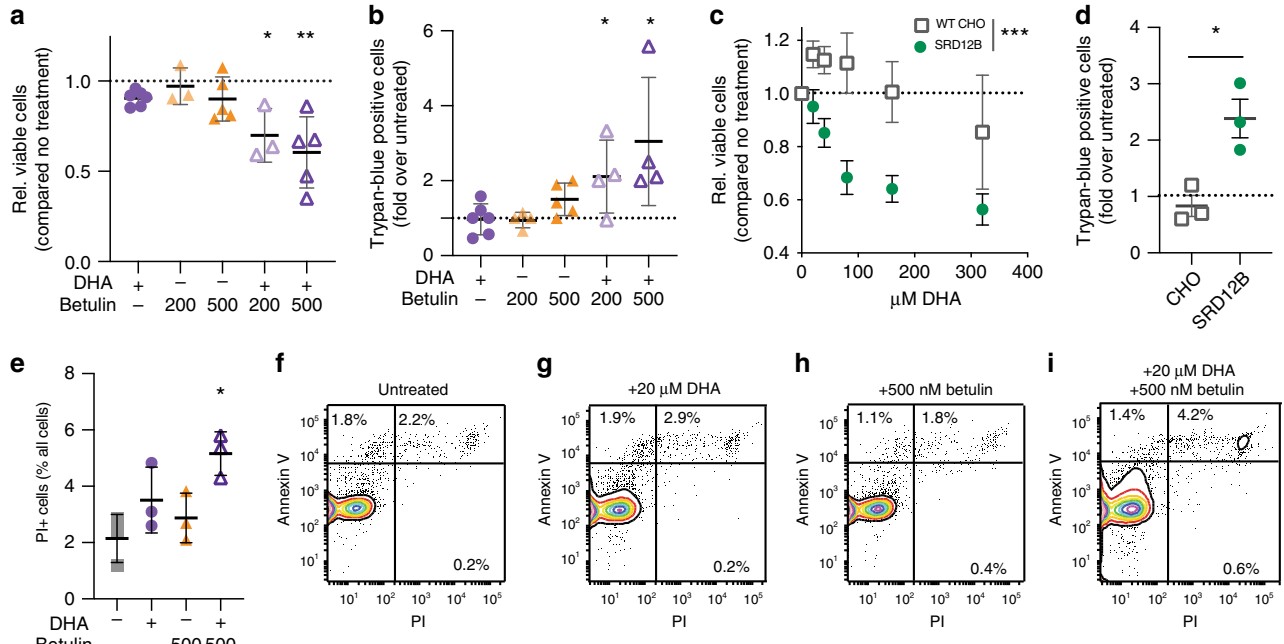

**Fig 6 Inhibition of homeostatic lipidome remodeling induces non-apoptotic cytotoxicity. a** Neither DHA nor betulin alone were cytotoxic in RBLs, whereas their combination significantly reduced viability. Full dose-response in Supplementary Fig. 10a **b** 24 h of DHA + betulin treatment significantly increased trypan-positive cells. **c** In SRD-12B cells, which fail to upregulate cholesterol and saturated lipids upon DHA treatment (see Fig. 4d and Supp Fig. S6), DHA significantly inhibited cell growth, in contrast to control WT CHO cells. **d** In SRD-12B cells, DHA significantly increased the proportion of TB+ cells, in contrast to WT CHO cells. **e** DHA + betulin treatment for 24 h significantly increased the proportion of PI+ cells. **f–i** Flow cytometry showed no increase in AnxV+/PI− cells (reflective of apoptosis), while the AnxV+/PI+ cells (non-apoptotic cell death) were increased. Flow cytometry positive controls shown in Fig. S13. All data shown as mean ± SD for ≥3 independent experiments. (**a**), (**b**), and (**d**) are one-sample $t$ tests compared to untreated; (**e**) is student's $t$ test compared to untreated; (**c**) is two-way ANOVA for effect of treatment. *$p < 0.05$, **$p < 0.01$, ***$p < 0.001$. Source data are provided as a Source Data file.

through activation of SREBP2 (Fig. 4). Disruption of SREBP activation resulted in inhibition of the compensatory response, both with respect to lipid composition (Figs. 3, 4, and Supplementary Fig. 7) and biophysical properties (Fig. 5). The implication of SREBP2 as a node of the sense-and-respond module for membrane adaptation is consistent with the central role of SREBPs in membrane homeostasis[43]. SREBP transcription factors have been dubbed the "master regulators of lipid homeostasis", because they direct not only cholesterol synthesis and uptake, but also proteins associated with membrane lipid metabolism, including those involved in FA synthesis, elongation, and desaturation[43]. The effects of PUFA supplementation on SREBP signaling have been extensively studied, with DHA known to suppress the activation of SREBP1 and its target genes, both in cultured cells and in vivo[62,63]. In contrast, SREBP2 is not suppressed by PUFAs[62,63], revealing that these two complementary regulators of membrane homeostasis have different functions, despite both being sensitive to membrane cholesterol. In our observations, SREBP2 is proteolytically activated as part of the homeostatic response. Previous reports[63] have not noted a significant effect of PUFAs on SREBP2, possibly because those experiments involved acute PUFA feeding of serum-starved cells, where high levels of activated SREBPs may have suppressed the DHA-mediated stimulation we observe. We have confirmed the possibility for such an effect in our cells (Supplementary Fig. 14). It remains to be determined whether SREBP2 is simply an effector of membrane remodeling downstream of yet-unidentified sensing machinery, or whether this protein (and/or its regulatory machinery) may itself be capable of sensing perturbations in membrane physical properties. Direct demonstration of protein responsiveness to membrane packing has recently been described

for two yeast ER proteins[5,14], providing a conceptual and methodological toolbox for identifying other membrane sensors. Remarkably, despite the ubiquity and importance of membrane homeostasis, the machinery used for sensing membrane properties remains largely uncharacterized. It is an intriguing observation that Ire1, a core component of the unfolded protein response (UPR), exhibits a dual sensitivity to unfolded proteins and aberrant lipid compositions, demonstrating a tight connection between protein-folding and membrane properties[14]. This connection suggests that the machinery for ameliorating protein-folding stresses may also be involved in transducing and mitigating membrane stress[64].

With extensive literature on the effects of dietary lipids on membrane compositions, it is somewhat surprising that the remodeling we observe has not been previously reported. We note that in many of the published datasets, increases in saturated fatty acids have indeed been observed, though rarely remarked upon. We speculate that the compensatory effects may have been previously overlooked for several reasons: (a) remodeling effects are obviously less pronounced than the "main" effects (incorporation and substitution) and may have been considered incidental/irrelevant; (b) lipid remodeling is temporally regulated and the time point of sample collection may affect its magnitude;[65] (c) most previous reports have analyzed isolated FAs rather than intact lipids, which may obscure how these are combined into phospholipids (e.g. it is possible to have more fully saturated lipids without a major increase in saturated FAs); (d) remodeling of the non-supplemented FAs can be obscured by the high abundance of the supplemented FAs. For example, if a large fraction of the phospholipidome is converted into DHA-containing lipids by DHA feeding, then the mol fraction of other lipids obviously

decreases, even if the relative proportion of saturated acyl chains compared to unsaturated is increased; (e) cholesterol is rarely considered alongside FAs and is a major component of the remodeling we report.

In summary, our observations support the hypothesis that mammalian cells in culture and in vivo sense membrane physical properties and respond to perturbations by comprehensive remodeling of their lipidomes. In our measurements, the perturbations were induced by supplementation with PUFAs; however, it is likely that other fats (e.g. cholesterol) or amphiphiles (e.g. bile acids, anesthetics) may induce similar responses. We show that the compensatory response is necessary for cell fitness, confirming membrane adaptation as a central requirement for cellular homeostasis.

## Methods

**Materials.** DHA, AA, OA, PA, betulin, GSK2194069, PF-429242, fluorescein diacetate, and 1,10 phenanthroline were obtained from Sigma Aldrich. C-Laurdan was purchased from TPProbes (South Korea). Amplex Red kit to quantify cholesterol was purchased from Invitrogen. Antibodies used: "actin (1:5000, monoclonal clone AC-15, Abcam), SREBP2 (1:400, polyclonal, Abcam), SREBP1 (1:1000, monoclonal, Abcam), ATF6 mentioned (1:1000, polyclonal, Bethyl Laboratories). Di-4-ANEPPDHQ (Di4), CellEvent™ Caspase-3/7 Green Detection Reagent, Annexin V-Pacific Blue, and Hoechst 33342 purchased from ThermoFisher. Fluorescein sodium salt was purchased from Santa Cruz Biotechnology.

**Cell culture.** Rat basophilic leukemia (RBL-2H3) cells (ATCC, CRL-2256), were maintained in medium containing 60% modified Eagle's medium (MEM), 30% RPMI, 10% fetal calf serum, 100 units/mL penicillin, and 100 µg/mL streptomycin. CHO cells, kindly provided by Anne Robinson (Carnegie Mellon University), were maintained in DMEM:F12 (1:1) containing 5% fetal calf serum, 100 units/mL penicillin, and 100 µg/mL streptomycin. SRD12B cells, kindly provided by Anne Robinson (Carnegie Mellon University), were maintained in DMEM:F12 (1:1) containing 5% fetal calf serum, 50 µM sodium mevalonate, 20 µM oleic acid, 5 µg/mL cholesterol, 100 units/mL penicillin, and 100 µg/mL streptomycin, as previously described[66]. All cells were grown at 37 °C in humidified 5% $CO_2$.

**Fatty acid treatments.** Fatty acid stock solutions were received as air-purged ampules (Sigma Aldrich) and loaded into BSA immediately upon opening. BSA loading was accompanied by stirring the fatty acid with BSA dissolved in water (2:1 mol/mol FA:BSA), sterile filtering, purging with nitrogen prior to aliquoting, and storing at −80 °C. BSA loading, purging, and cold storage were all done to minimize FA oxidation. For all experiments with FA supplementation, cells were incubated with 20 µM FA for 3 days (cells were re-supplemented 24 h prior to analysis). Serum-containing, rather than serum-free, media was used to approximate physiological supplementation, rather than exclusive introduction of specific FAs. These conditions were chosen to approximate in vivo settings produced by DHA-enriched diets in mammals: plasma free FA concentrations range from 300 to 750 µM[67,68] and up to 10 mol % of plasma fatty acids are ω-3 DHA in rats fed a high-fish-oil diet[67]. Further, diets rich in ω-3 PUFAs led to significant incorporation of these fats into cell membrane lipids[15,32,67], similar to the levels we observed under our culture feeding conditions (Fig. 1). For these reasons, we believe our culture conditions reasonably approximate physiological dietary membrane perturbations.

**Animals and diets.** All experimental procedures using laboratory animals were approved by the Public Health Service and the Institutional Animal Care and Use Committee at Texas A&M University. Pathogen-free female C57BL/6 mice (n = 8) weighing 16–18 g, were randomly divided into two groups of 4. For 2 weeks, mice had free access to one of the two semi-purified diets, which were adequate in all nutrients[30]. Diets varied only in the oil composition, i.e., either corn oil (CO) or a PUFA-enriched fish-corn oil (FO) mixture (4:1, w/w) at 5 g/100 g diet. The basic diet composition, expressed as g/100 g was: casein, 20; sucrose, 42; cornstarch, 22; cellulose, 6; AIN-76 mineral mix, 3.5; AIN-76 vitamin mix, 1, DL-methionine, 0.3; choline chloride, 0.2; Tenox 20 A, 0.1; and oil, 5. The fatty acid composition of the diets was determined by gas chromatography (Supplementary Table 1). Mice were sacrificed by CO2 asphyxiation after 2 weeks of feeding the CO/FO diet, and the livers and hearts were frozen in liquid nitrogen. 20–50 mg of tissue was mechanically homogenized in Dulbecco's PBS (without $Ca^{2+}$ and $Mg^{2+}$). Samples were further diluted to 5 mg/mL in Dulbecco's PBS (without $Ca^{2+}$ and $Mg^{2+}$), and lipidomics analysis was performed as below.

**Drug treatments and cell number quantification.** RBL cells were treated with or without 20 µM DHA in the presence or absence of betulin (inhibitor of SREBP processing) or GSK2194069 (FAS inhibitor). Cells were treated for 72 h, and then

the cell number determined via fluorescein diacetate (FDA) fluorescence. FDA is a cell viability probe which freely diffuses through cell membranes but is trapped in cells following de-acytelation by cytoplasmic esterases in viable cells. The number of viable cells is then directly related to fluorescein fluorescence intensity. For the assay, cells were treated in 96-well plates, gently washed with PBS, and then incubated with fluorescein diacetate (5 µg/mL in PBS) for 2 min at 37 °C. The plates were then washed again to remove excess FDA and fluorescence was measured at 488 nm excitation and 520 nm emission. For caspase inhibitor studies, 5 and 30 µM Z-DEVD-FMK were added at the same time as DHA and betulin and treated for 72 h prior to FDA analysis.

To count the number of dead cells, cells were treated for 24 h, and the supernatant harvested. The cells were then trypsinized and combined with the supernatant. 0.4% Trypan Blue was added to an equal volume of cells. The cells were then counted using a Countess II cell counter (Thermo Fisher), and the number of Trypan-positive cells recorded. For each experiment the number of cells per well were normalized to untreated, unsupplemented cells.

**Active caspase quantification.** RBL cells were treated with or without 20 µM DHA in the presence or absence of 500 nM betulin for 24 h. After 24 h, the supernatant was removed, and CellEvent™ Caspase-3/7 Green Detection Reagent (7.5 µM, diluted in PBS + 0.5% FBS) was added to each well. The cells were incubated at 37 °C for 30 min. At this time Hoechst 33342 (2.5µg/mL final) was added to the solution and incubated for an addition 5 min at 37 °C. Two images using DAPI and GFP filters were randomly taken for each condition in three different random locations in the well. For a positive control, cells were treated with 2 mM N-ethylmaleimde in 50 mM HEPES, 2 mM CaCl2, pH 7.4 for 45 min prior to addition of the caspase detection reagent. Using ImageJ, to determine positive staining for caspase activity, a threshold was set using the positive controls, and this threshold was maintained across the experiment. The number of caspase-positive cells was normalized to the number of cells imaged in the same field of view by counting the Hoechst-positive nuclei.

**Lipidomics.** All lipidomic datasets, with species abundance given in picomoles, are available in Supplementary Data 1. Detailed lipidomic analysis was performed by Lipotype, GmbH, as previously described[69]. Briefly, for preparation of crude cell membranes, cells were washed with phosphate-buffered saline (PBS), scraped in 10 mM Tris pH 7.4, and then homogenized with a 27-gauge needle. Nuclei were then pelleted by centrifugation at 300$g$ for 5 min. The supernatant was pelleted by centrifugation at 100,000$g$ for 1 h at 4 ºC. The membrane pellet was then washed and resuspended in 150 mM ammonium bicarbonate.

All lipidomics were performed at Lipotype GmbH (Dresden, Germany) as described previously[17–19,22,28,70] and detailed below.

The following lipid names and abbreviations are used: ceramide (Cer), Chol, SM, diacylglycerol (DAG), lactosyl ceramide (DiHexCer), glucosyl/galactosyl ceramide (HexCer), sterol ester (SE), and triacylglycerol (TAG), as well as phosphatidic acid (PA), phosphatidylcholine (PC), phosphatidylethanolamine (PE), phosphatidylglycerol (PG), and phosphatidylinositol (PI), phosphatidylserine (PS), and their respective lysospecies (lysoPA, lysoPC, lysoPE, lysoPI, and lysoPS) and ether derivatives (PC O-, PE O-, LPC O-, and LPE O-). Lipid species were annotated according to their molecular composition as follows: [lipid class]-[sum of carbon atoms in the FAs]:[sum of double bonds in the FAs];[sum of hydroxyl groups in the long chain base and the FA moiety] (e.g., SM-32:2;1). Where available, the individual FA composition according to the same rule is given in brackets (e.g., 18:1;0-24:2;0).

Synthetic lipid standards were purchased from Sigma-Aldrich (Chol D6), Larodan (Solna, Sweden) Fine Chemicals (DAG and TAG), and Avanti Polar Lipids (all others).

Lipid extraction for mass spectrometry lipidomics: Lipids were extracted using a two-step chloroform/methanol procedure[22]. Samples were spiked with internal lipid standard mixture containing: cardiolipin 16:1/15:0/15:0/15:0 (CL), ceramide 18:1;2/17:0 (Cer), diacylglycerol 17:0/17:0 (DAG), hexosylceramide 18:1;2/12:0 (HexCer), lyso-phosphatidate 17:0 (LPA), lyso-phosphatidylcholine 12:0 (LPC), lyso-phosphatidylethanolamine 17:1 (LPE), lyso-phosphatidylglycerol 17:1 (LPG), lyso-phosphatidylinositol 17:1 (LPI), lyso-phosphatidylserine 17:1 (LPS), phosphatidate 17:0/17:0 (PA), phosphatidylcholine 17:0/17:0 (PC), phosphatidylethanolamine 17:0/17:0 (PE), phosphatidylglycerol 17:0/17:0 (PG), phosphatidylinositol 16:0/16:0 (PI), phosphatidylserine 17:0/17:0 (PS), cholesterol ester 20:0 (CE), sphingomyelin 18:1;2/12:0;0 (SM), triacylglycerol 17:0/17:0/17:0 (TAG) and cholesterol D6 (Chol). After extraction, the organic phase was transferred to an infusion plate and dried in a speed vacuum concentrator. 1st step dry extract was resuspended in 7.5 mM ammonium acetate in chloroform/methanol/propanol (1:2:4, V:V:V) and 2nd step dry extract in 33% ethanol solution of methylamine in chloroform/methanol (0.003:5:1; V:V:V). All liquid handling steps were performed using Hamilton Robotics STARlet robotic platform with the Anti Droplet Control feature for organic solvents pipetting.

MS data acquisition: Samples were analyzed by direct infusion on a QExactive mass spectrometer (Thermo Scientific) equipped with a TriVersa NanoMate ion source (Advion Biosciences). Samples were analyzed in both positive and negative ion modes with a resolution of $R_{m/z=200} = 280,000$ for MS and $R_{m/z=200} = 17,500$ for MS/MS experiments, in a single acquisition. MS/MS was triggered by an inclusion

list encompassing corresponding MS mass ranges scanned in 1 Da increments[70]. Both MS and MS/MS data were combined to monitor CE, DAG and TAG ions as ammonium adducts; PC, PC O-, as acetate adducts; and CL, PA, PE, PE O-, PG, PI and PS as deprotonated anions. MS only was used to monitor LPA, LPE, LPE O-, LPI and LPS as deprotonated anions; Cer, HexCer, SM, LPC and LPC O- as acetate adducts and cholesterol as ammonium adduct of an acetylated derivative[71].

Data were analyzed with in-house developed lipid identification software based on LipidXplorer (20). Data post-processing and normalization were performed using an in-house developed data management system. Only lipid identifications with a signal-to-noise ratio >5, and a signal intensity 5-fold higher than in corresponding blank samples were considered for further data analysis.

The lipidomic analysis yields a list of >600 individual lipid species and their picomolar abundances. These were processed by first transforming into mol% of all lipids detected. Next, the TAG and sterol esters were removed from the analysis, and the remaining data was analyzed as mol% of membrane lipids. From here, the datasets were broken down further into class composition. In some cases, the distribution and structural characteristics (e.g. number of carbons or unsaturations in the acyl chains) of the individual species were analyzed.

Both DHA and AA can be converted into other PUFAs. DHA can be retro-converted to EPA (20:5) which can then be converted to 24:6. AA can be converted into 22:4 and 24:4. In our supplemented cells, we see small, but non-negligible increases in lipids containing these converted, supplemented fatty acyl chains. With DHA supplementation, the retroconversion represented <4.8 mol% GPLs. With AA supplementation, lipids containing converted FAs range from 13–15%. Thus, ω−3 PUFAs are defind here (Fig. 1) as lipids containing either 22:6, 20:5, or 24:6 as an acyl chain. ω−6 PUFAs are lipids which contain 20:4, 22:4, or 24:4 as an acyl chain.

**C-laurdan spectroscopy**. Membrane packing (related to order and fluidity) was determined via C-laurdan spectroscopy as described[72,73]. Briefly, cells were washed with PBS and stained with 20 μg/mL C-laurdan for 10 min on ice. The emission spectrum from isolated GPMVs was gathered from 400–550 nm with excitation at 385 nm at 23 °C. The GP was calculated according to the following equation:

$$GP = \frac{\sum_{420}^{460} I_x - \sum_{470}^{510} I_x}{\sum_{420}^{460} I_x + \sum_{470}^{510} I_x}. \quad (1)$$

**C-Laurdan spectral imaging**. C-Laurdan imaging was performed as previously described[19,28,52,72,73]. Briefly, cells were washed with PBS and stained with 10 μg/mL C-Laurdan for 10 min on ice, then imaged via confocal microscopy on a Nikon A1R with spectral imaging at 60× and excitation at 405 nm. The emission was collected in two bands: 433–463 nm and 473–503 nm. MATLAB (MathWorks, Natick, MA) was used to calculate the two-dimensional (2D) GP map, where GP for each pixel was calculated from a ratio of the two fluorescence channels, as previously described[52]. Briefly, each image was binned (2 × 2), background subtracted, and thresholded to keep only pixels with intensities greater than 3 standard deviations of the background value in both channels. Supplementary Fig. 15 shows representative images of each of these image processing steps. The GP image was calculated for each pixel using the above equation. GP maps (pixels represented by GP value rather than intensity) were imported into ImageJ. To ensure that the specifics of our quantification method were not biasing the result, we quantified the GP of internal membranes by three different methods. First, line scans were drawn across individual cells and PM GP values were taken as peak GP values from the periphery of the cell, whereas internal membranes were calculated as the average of all values outside the PM peak. Alternatively, the nucleus, visible as a dark spot in C-Laurdan images, was used as a fiducial marker to draw a region-of-interest representing peri-nuclear membranes. The average GP of those pixels was taken as representative of internal membranes. Finally, as shown in the images in Fig. 5, the internal membranes were masked, and histograms of the pixels inside the mask were plotted. Ultimately, all three methods produced quantitatively similar results, suggesting that this measurement is robust. We chose to present the results of the third method here because it includes the most information, allowing us to evaluate the distributions of pixels GPs.

**Di4 fluorescence lifetime imaging**. RBL cells were treated for 24 h prior to staining and imaging. Outer leaflet staining of RBL cell PMs was achieved by incubating cells at 4 °C for 8 min with Di4 at 1 μg/ml in phosphate-buffered saline. Cells were quickly washed prior to immediate imaging in phenol red-free MEM. Lifetime microscopy was performed on a Nikon A1 laser scanning microscope with an integrated Picoquant time-correlated single photon counting (TC-SPC) system (Berlin, Germany). The instrument response function (IRF) was determined with a saturated erythrosine B and KI solution at pH=10 according to the manufacturer's (Picoquant) protocol. Di4 emission was collected at >560 nm, and images were acquired using 20 MHz pulse frequency. The photon count rate was kept under 10% of the pulse rate by adjusting a manual shutter, and enough frames were acquired to obtain at least $8 \times 10^3$ photons cumulative signal intensity. The fluorescence decay curves were fitted to a bi-exponential re-convolution function adjusted to the IRF, and the average lifetime was calculated and represented in the FLIM images as $\tau_{Di4}$. The experiment was repeated four times, imaging 15–30 cells per condition each time.

**Amplex Red assay**. Amplex Red cholesterol assay was performed (according to manufacturer instructions; Invitrogen) to determine the abundance of cholesterol. Each reading was normalized to protein concentration (determined by bicinchoninic acid (BCA) assay) in the same samples. Technical triplicates were measured for each sample. Shown are values normalized to untreated cells. Reported are the average and standard deviations from at least three independent biological replicates.

**Western Blot**. Cells were washed with ice cold PBS, then scraped into Laemmli lysis buffer (50 mM Tris-HCl, pH 8.0; 2% SDS; 5 mM EDTA, pH8.0) supplemented with protease inhibitor cocktail. Protein concentration was determined using BCA (Pierce), and equal amounts of protein were mixed with reducing Laemmli sample buffer and loaded onto SDS-PAGE gels. Gels were transferred to PVDF membranes, which were blocked in 5% BSA. Membranes were incubated with primary antibodies overnight at 4 °C, and detected with either AlexaFluor or HRP-tagged secondary antibodies. Membranes were imaged using a BioRad ChemiDoc imager. The intensities of the bands were quantified normalized to actin, and plotted as mean ± SD of $n \geq 3$ experiments.

**Real time quantitative PCR**. Total RNA was isolated via Trizol (Sigma) following the manufacturer's protocol. Reverse transcriptase PCR was performed using the High Capacity cDNA Reverse Transcription Kit from Applied Biosystems according to manufacturer's protocol. To quantify mRNA expression, SYBR Fast MasterMix (2×) Universal Dye (#KK4602) from Kapa Biosystems was used in an Eppendorf Realplex2 Mastercycler. Each primer set for each sample was run in triplicate with 1 ng of cDNA per well. The primer sets used are included as Supplementary Table 2. Expression changes were calculated using the delta delta $C_T$ method.

**Permeability assay**. RBL cells were treated with or without 20 μM DHA in the presence or absence of 500 nM betulin for 24 h. At this time, cells were trypsinized, washed 1x with full serum-containing medium and 2× with phenol red-free MEM, and counted with a Countess II automated cell counter (ThermoFisher). 250,000 cells were diluted in 2 mL with phenol red-free MEM and added to a quartz 2 mL cuvette. Just prior to reading on the spectrofluorometer, 2.5 ug/mL fluorescein diacetate (stock 1 mg/mL in acetone) was added to the cells and mixed by pipetting. The fluorescence intensity was read at 450 nm excitation and 510 nm emission over a period of 5 min (1 read/second). In each of the samples measured, the slope of the fluorescence increase over time remained constant over the entire 5 min, consistent with a constant flux $(Q)$.

The permeability coefficient $(P)$ of a membrane to a particular substrate is then defined by Fick's Law as

$$Q = P * A * (C_{out} - C_{in}), \quad (2)$$

where $A$ is the area of the membrane and $C_{out}$ and $C_{in}$ are the concentrations of the solute outside and inside the cells, respectively. In our experiments, $C_{in} = 0$ because FDA is quickly converted into a different molecule upon entering the cell. We estimate the transport surface area by multiplying a typical mammalian cell surface area (~3000 μm$^2$) by the number of cells in the cuvette (250,000). Using pure fluorescein for calibration, we measured directly the flux of FDA molecules across the membrane, which for a typical sample of untreated RBL cells was $6.6 \times 10^{-5}$ nmol/s. From these parameters, we calculated a permeability coefficient for FDA to be ~$2.2 \times 10^{-6}$ cm/s, in excellent agreement with values previously reported in plant cells[59]. We found that the flux scales linearly with cell and FDA concentrations, consistent with the simple Fick's Law model of passive diffusion followed by rapid active conversion.

**Reporting summary**. Further information on research design is available in the Nature Research Reporting Summary linked to this article.

## Data availability

Data supporting the findings of this manuscript are available from the corresponding authors upon reasonable request. A reporting summary for this Article is available as a Supplementary Information file. The source data underlying all figures are provided as a Source Data file. The raw lipidomes (in picomoles of lipid species) are provided in Supplementary spreadsheets.

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

## Acknowledgements

All fluorescence microscopy was performed at the Center for Advanced Microscopy, Department of Integrative Biology & Pharmacology at McGovern Medical School, UTHealth. We thank Neal Waxham for providing rat hippocampal neurons. We gratefully acknowledge Theodore Steck, Yvonne Lange, and members of the Levental lab for their critical feedback on this manuscript. Funding for this work was provided by: the NIH/National Institute of General Medical Sciences (1R01GM114282, 1R01GM124072, 1R35GM134949), the Volkswagen Foundation (grant 93091), and the Human Frontiers Science Program (RGP0059/2019) to IL; NIH/National Institute of General Medical Sciences 1R0GM120351 to KL; T32 GM 008280-31to JLS; NIH/National Cancer Institute R35-CA197707 and funds from the Allen Endowed Chair in Nutrition & Chronic Disease Prevention to RSC.

## Author contributions

K.R.L. and I.L. conceived the idea and designed the experiments. K.R.L., E.M., J.L.S., and Y.F. performed the experiments and analyzed the data. K.R.L., R.S.C., and I.L. supervised the research. K.R.L. and I.L. wrote the manuscript with input from R.E.

## Competing interests

The authors declare no competing interests.
