## [Peer Review File · Nature Communications]

Reviewers' Comments:

Reviewer #1:

Remarks to the Author:

The manuscript entitled "Homeostatic remodeling of mammalian membranes in response to dietary lipids is essential for cellular fitness" from Levental and collaborators demonstrates that homoviscous adaptation occurs in mammalian cell lines as well as in a mouse model. The authors propose that enriching cell membranes with polyunsaturated fatty acids (PUFAs) upregulates the production of cholesterol and saturated fatty acids. This in turn counterbalances the effect that PUFAs exert on membrane fluidity. Most of the conclusions are logically driven from their experimental results and from previous evidence, which the authors properly cited and put in context.

I truly believe that the manuscript from Levental et al advances our knowledge on the mechanism behind membrane remodeling upon enrichment with dietary fatty acids. This is relevant not only for researchers working with lipids and metabolism, but also for researchers that study membrane protein function and regulation by lipids.

Please, find some comments below which I think would improve the paper:

1- The authors state on the Introduction that membrane homeostatic responses driven by dietary fatty acids are centrally mediated by SREBP2. Although this reviewer agrees that SREBP2 plays a crucial role in lipid remodeling, it is not clear whether other proteins in the same (or other) pathway might also play a role. For readers outside this field, it would be a good idea to introduce SREBP2 at the beginning of the manuscript and its role in lipid metabolism, particularly why it was chosen over other protein candidates. The authors should consider testing other (1 or 2) proteins associated with lipid metabolism for which there is no homeostatic effect after PUFAs treatment.

2- How were toxicity or differences in proliferation measured? (lines 141-142).

3- I encourage the authors to show GP maps of cells treated with DHA at different time points after supplementation. It would be illustrative to compare with the control and let the readers appreciate the differences in the raw data.

Minor comments:

1- On the last paragraph of the Introduction (lines 91-100), it would be more informative if the authors add the key experimental approaches when summarizing the experimental findings that drove the conclusion. This would guide the reader since the beginning of the manuscript.

2- Authors need to format references on line 218.

3- Figure 3 would be clearer if it included an inset (or scale break) highlighting the section between 0 and 18 hours to properly assess the simultaneous compensatory effect.

4- Legends should reveal what fits were used to adjust to the data.

5- Rephrase "if this lipidome response were designed to..." (line 231), as it seems to point out to intelligent design.

6- For clarity and reproducibility, include the whole western blots in figures 4D and S8.

Reviewer #2:

Remarks to the Author:

Summary

In this study Levental et al. investigate homeostatic adaptation in mammalian cells, both at the cell and organ level using a combination of approaches – lipidomics, microscopy and Western blotting. They find that exposing cells to PUFAs results in their quick incorporation in membrane lipids, a change that within an hour is compensated by an increase in the level of saturated fatty acids and cholesterol as well as a decrease in other PUFAs than the ones supplemented. Similar changes in phospholipids are observed in mouse heart and liver. The transcription factor SREBP2 is implicated in the membrane lipid composition response in that its mature form is upregulated upon DHA-treatment, that the cholesterol increase is not seen in SRD-12B cells where it can not mature and after inhibition of SREBP2 processing by betulin. The study is mostly sound and the authors are correct in that not much is known about lipid remodeling in mammalian cells. However, little of what is presented in this paper is new and the increased knowledge is incremental rather than substantial.

Major points

1. There is no reason to speculate about the acyl chain composition in cell culture media. Using mass spectrometry lipidomics, it should be possible to estimate the levels of the supplemented fatty acids in the cell culture media and state the level of supplementation.
2. In Figure 4 the mature form of SREBP2 gives a double band but in Figure S8 there is a single band for the same protein. Please provide the entire Western blot to confirm that the correct bands are shown and explain the discrepancy. Moreover it is stated that there was no effect in the level of the precursor form of SREBP2, but it appears that the actin band (presumably the loading control) decreases in intensity at the longer times suggesting that there is a factual increase in the precursor form. Please quantify and discuss the results.
3. Figure 5 raises a lot of questions:
 - a) The effect of fatty acid supplementation on the GP-values in the plasma membrane are neither reported nor commented upon. Presumably because no change could be detected. Please report the results and discuss them.
 - b) It is suggested in the main text, but not in the figure legend that panel C corresponds to 24h. Would it not then be the same as the last time point in panel B? However, the mean values and the spreads are not identical. Please explain why that is and state whether the GP-value in the +DHA and +betulin group is statistically different from the +DHA-group.
 - c) Please clarify what fraction of the total of the intracellular pixels were included in analysis after discarding all pixels with intensities lower than 3 standard deviations of the background value after the binning process. (Material and Methods page 21, lines 510-511).
 - d) In panels A and B, please indicate what parts of the line scan was used for calculating the GP-values of the internal membranes.
4. For Figures S4 and S5 statistics need to be supplied in order to allow any statements on decreases in gene expression and lipid saturation.
5. The data in Figure S6 suggests that betulin treatment increases the fraction of saturated lipids. Please provide the statistics and comment upon the results.

Minor points

1. In Figure 6, the y-axes of the A- and B-panels are labeled incorrectly.
2. Figure S1. Please clarify if the data are new or copies from the 2017 paper.

3. In Figure S2 panels A and B the labelling of +PA and +OA does not match the figure legend.
4. In Figure S7, something odd has happened with the last data point and there appears to be both a mix-up of the symbols and a missing data point.
5. Please comment upon why SRD12B cells by default were maintained in media supplemented with mevalonate, oleic acid and cholesterol. (Page 18 lines 407-408).
6. Please explain why the diet contained glycerol and propylene glycol. (Page 18, lines 430-431).
7. Line 349 should read temperature rather than temperatures.
8. Line 465 should read 4°C rather than 4_C

Reviewer #3:

Remarks to the Author:

Overview:

“Homeoviscous adaptation” is the preservation of membrane properties despite a changing external environment. Changes in membrane lipid composition in response to temperature fluctuations are well characterized in ectothermic species, but processes governing mammalian membrane homeostasis are less studied, as mammals maintain a somewhat constant internal body temperature. Here the authors provide experimental evidence showing that mammals actively modulate their lipidomes in response to changes in dietary lipid. In general, highly unsaturated omega-3 and omega-6 fatty acids from the diet are preferentially incorporated into membrane lipids, and the authors demonstrate that cells respond by increasing incorporation of cholesterol and unsaturated lipids into membrane lipids as well. The authors claim that this is the first demonstration of homeoviscous adaptation in response to dietary lipids in mammals, and thus has important implications for the understanding of mammalian membrane homeostasis. The authors seek to demonstrate that (i) dietary PUFAs are preferentially incorporated into membrane lipids, (ii) in response, cells actively incorporate stabilizing lipids into their membranes, and (iii) these changes are essential for cellular fitness. The first point is well supported in the existing literature, so these experiments are important controls but not novel findings. The second point is both novel and well-supported by the data, showing that lipidomes are remodeled following PUFA incorporation, but mechanistic insight is relatively shallow (it is shown to be SREBP-dependent). Regarding the third point, cell growth in culture was the only metric of cellular fitness reported, and is therefore not sufficient to be broadly indicative of cellular fitness. As it is written, the claims and conclusions of the paper exceed what is supported by the data and appear more suited for a more specialized journal

Major concerns:

1. In the title, as well as throughout the text, the authors state that lipidome remodeling is essential for “cellular fitness”. However, the only assay used to evaluate cellular fitness is growth rate (relative cell number after 3 days, Fig. 6C). In order to make claims about cellular fitness, the authors would need to demonstrate:
 - a. Genetic or cytological indicators of reduced cellular fitness or increased cellular stress (independent of growth rate). These are alluded to in lines 99-100 but never performed/reported (“leads to cytotoxic effects”).
 - b. Extend these findings to demonstrate cytotoxicity and/or show some kind of physiological response in vivo when remodeling is blocked
 - c. Eliminate the possibility that DHA/lipidome remodeling directly modulates rates of cell division (e.g. via a mitogenic pathway, rather than indirectly through changes in cellular fitness). Direct

modulation of division rates seems particularly probable given that low-dose DHA appears to stimulate cell division in WT cells (see fig. 6C), contrary to statements made in the results section (line 141-142, stating that no "obvious ... differences in proliferation" were observed).

Without addressing the above points, it does not seem appropriate to make any claims about cellular fitness, but rather limit discussion to the impact of DHA/lipidome remodeling to effects on growth rate of cultured cells.

2. Data are not quantified correctly. Figures 1 A, B, D, and E have Y-axes that measure a particular fatty acid based on chain length and degree of unsaturation (22:6 for DHA supplementation experiments, and 20:4 for AA supplementation). However, Supplemental figure 3A shows that both AA and DHA are desaturated significantly prior to incorporation into glycerophospholipids (for example ~50% of the DHA appears to be incorporated as 22:6, but the remaining 50% of incorporated DHA appears to be in the form of fatty acids with >6 degrees of unsaturation). Therefore, it may be more appropriate to quantify changes in omega-3 and omega-6 fatty acids, rather than changes in 22:6 and 20:4 fatty acids. Ideally, radiolabeled fatty acids could be used to directly track the incorporation of fatty acids into different species even after remodeling.

3. The discussion of in vivo experiments is misleading, and the experimental design is questionable. Interpretation of the in vitro studies leads readers to understand that dietary omega-3 and omega-6 PUFAs are "robustly and specifically incorporated ... into membrane lipids" (section 1 of the results), which leads to a concomitant increase in unsaturated lipids to maintain membrane homeostasis. Then, in vivo experiments compare fish oil supplementation to corn oil supplementation. In the main text, corn oil is described as a poor source of omega-3 PUFAs (line 122). However, corn oil is a very rich source of omega-6 PUFAs in the form of linoleic acid (18:2). If readers were told that corn oil is a rich source of Omega-6 PUFAs, they might expect it to induce a similar lipidome remodeling as seen in fish oil, but strangely such an effect is not observed (Fig. 2D). This is an interesting result, which seems to indicate that highly unsaturated omega-3/6 fatty acids (DHA and AA) induce lipidome remodeling, whereas the more saturated omega-6 PUFA (LA) does not. The main text needs to be modified to highlight the omega-6 PUFA content of corn oil, and the resulting discussion needs to be significantly restructured to emphasize differences between moderately and highly unsaturated PUFAs. Additional experiments may need to be performed if the authors wish to demonstrate differences in lipidome remodeling in response to level of unsaturation.

4. SREBP experiments are incomplete. Only the SREBP2 isoform was studied (fig. 4C), however the experimental manipulations performed (betulin and SRD12B) interfere with SREBP1(a and c) as well. Moreover, SRD12b disrupts all transcription factors that are golgi processed by S1P (e.g. CREBH, ATF6...) not just SREBP. This is important because these other transcription factors have known effects on ER biology, lipid metabolism and trafficking (see Kim et al. Mol Cell Biol. 2017 Jun 29;37(14) and Cheng et al. J Biol Chem. 2016 Nov 4;291(45):23793-23803). Clarifying this point is particularly relevant given that disruption of SREBP processing causes a "failure to upregulate saturated lipids" (line 364), which is "not directly connected to SREBP2 target genes", but is directly connected to SREBP-1c target genes. Moreover, DHA supplementation has been frequently associated with suppression of SREBP1 and reduction in expression of desaturase genes (e.g. see Haung et al., BMC Cancer. 2017; 17: 890 and Deng et. al Biochim Biophys Acta. 2015 Dec;1851(12):1521-9 and Caputo J Cell Physiol. 2011 Jan;226(1):187-93). In sum, simply showing that SREBPs is involved is not enough given this extensive literature. Especially since the authors are arguing that DHA's upregulation of membrane cholesterol by SREBP2 is the key driver of the adaption.

5. There is a rich literature on PUFA supplementation in both cultured cells and in whole animals. Very few of these studies report the kind of changes in saturated fatty acids reported by the authors. For example, Harris et al report fatty acid data from people with heart biopsies following 6

months of w3 PUFA supplementation where they found no change in saturated fatty acids, an increase in w3 FAs (as expected due to the supplementation) and a reduction in w6 FAs (Circulation. 2004;110:1645–1649). Similarly, Owen et al., reported on rats fed from 0-42days a high w3 diet really did not see significant changes in myocardial saturated fatty acids. Yet, they did observe a major increase in w3 FAs and an associated reduction in w6 FAs (Lipids , 2004 Vol. 39, Iss. 10). In short, the authors discuss little of this vast literature of PUFA supplementation and how it does or does not agree with their findings.

6. Statistics are not reported for the majority of the supplemental figures, as well as several figures in the main text (Fig. 3, Fig. 4 C-D).

Minor concerns:

1. Supplemental figure 3A shows a significant decrease in monounsaturated lipids in response to AA treatment. It is unclear why this result is not mirrored in the corresponding figure in the main text (Fig. 2a), where monounsaturated lipids appear unchanged. This discrepancy needs to be discussed in the main text.
2. Lines 163 and 164 state that there was “a notable increase in relatively short non-DHA/AA containing lipids (Fig. S3D-E)”. However, no statistical test was described or reported to support this conclusion. It is therefore unclear whether there is a significant difference (in which case statistics need to be reported), or whether the authors wish to highlight a non-significant trend (which needs to be clearly qualified in the main text).
3. The authors claim that PUFAs were not incorporated as into TAGs, but this data is not shown. These data provide essential context for our interpretation of the degree of incorporation into GPLs.
4. There is insufficient discussion of membrane compartments. It is unclear why the GP is reported only for internal membranes (Fig. 5A), as it would be an interesting point of discussion if the plasma membrane composition responded differently than internal membranes.

We sincerely thank the editor and reviewers for their constructive critiques of our manuscript titled “*Homeostatic remodeling of mammalian membranes in response to dietary lipids is essential for cellular fitness*”. We have substantively revised the manuscript to address reviewer concerns, including 16 new experimental figure panels in the main text, 9 new supplementary figures, and extensive revision of the manuscript. Most notably, we have:

1. Carefully detailed the “cellular fitness” phenotype, showing that non-apoptotic cell death is responsible for the reduced cell numbers resulting from uncompensated membrane perturbations (Author response 3.2)
2. Analyzed the involvement of SREBP1, SREBP2, and other transcription factors in homeostatic membrane remodeling
3. In the course of these revisions, to explain the observed cytotoxicity phenotype, we have introduced a novel aspect to the manuscript by investigating the role of lipid perturbations on membrane permeability. To this end, we developed a novel methodology to quantify membrane permeability to amphiphilic substrates in living cells. We report that perturbed membranes are indeed more permeable and suggest that this increased permeability accounts for the observed cytotoxicity (Author response 3.3)

We believe that the reviewers’ comments, and the experimental evidence generated to address them, have dramatically improved the manuscript, and that it is now acceptable for publication in *Nature Communications*.

Below please find a point-by-point response to the reviewer comments.

Reviewer #1 comments:

The manuscript entitled “Homeostatic remodeling of mammalian membranes in response to dietary lipids is essential for cellular fitness” from Levental and collaborators demonstrates that homeoviscous adaptation occurs in mammalian cell lines as well as in a mouse model. The authors propose that enriching cell membranes with polyunsaturated fatty acids (PUFAs) upregulates the production of cholesterol and saturated fatty acids. This in turn counterbalances the effect that PUFAs exert on membrane fluidity. Most of the conclusions are logically driven from their experimental results and from previous evidence, which the authors properly cited and put in context.

I truly believe that the manuscript from Levental et al advances our knowledge on the mechanism behind membrane remodeling upon enrichment with dietary fatty acids. This is relevant not only for researchers working with lipids and metabolism, but also for researchers that study membrane protein function and regulation by lipids.

Authors’ response: We thank the reviewer for their detailed examination of our work and appreciate the positive feedback.

1- The authors state on the Introduction that membrane homeostatic responses driven by dietary fatty acids are centrally mediated by SREBP2. Although this reviewer agrees that SREBP2 plays a crucial role in lipid remodeling, it is not clear whether other proteins in the same (or other) pathway might also play a role. For readers outside this field, it would be a good idea to introduce SREBP2 at the beginning of the manuscript and its role in lipid metabolism, particularly why it was chosen over other protein candidates. The authors should consider testing other (1 or 2) proteins associated with lipid metabolism for which there is no homeostatic effect after PUFAs treatment.

Authors’ response 1.1: We agree with the reviewer. We did not appropriately justify our focus on SREBP2 in the original manuscript. Originally, we focused on this protein because of the observed upregulation of membrane cholesterol as part of the homeostatic response. Activation of SREBP2 is the major cellular mechanism for upregulation of cholesterol synthesis, thus we focused on this activation as a likely mediator of the homeostatic response. We have now added a discussion to the manuscript to support this line of reasoning.

Action taken - addition to Results section: “The machinery for cholesterol production in metazoans is regulated by proteolytic processing of transcription factors of the sterol regulatory element binding protein (SREBP) family. Specifically, signals to upregulate cellular cholesterol levels are translated into proteolysis of a membrane-bound SREBP2 precursor to release a ‘mature’ cleaved fragment, which translocates to the nucleus to induce transcription of various target genes, including those for cholesterol synthesis and uptake. Having observed a robust and rapid increase in cholesterol levels resulting from DHA supplementation, we evaluated whether SREBP2 processing was associated with this response.”

We have also added a supplemental figure (Fig S8) in which we tested inhibitors of various proteins involved in lipid metabolism and found that only inhibitors of SREBPs and FASN significantly affected homeostasis and cell fitness after PUFA treatment. Additionally, we used Western blotting and qPCR to analyze the effect of DHA supplementation on the

activation of ATF6 (processed by similar mechanisms as SREBPs) and the unfolded protein response sensor IRE1 (Fig S10). We observe no significant effect on ATF6 cleavage (via Western blotting) nor increase in ATF6- or UPR-induced gene expression (via qPCR). Finally, we have added a panel to Fig 4 in which we show that DHA does not affect SREBP1 proteolytic processing. We believe these results support our implication of SREBP2 as an important mediator of physical/lipidomic homeostasis. However, it is important to emphasize that we do not intend to claim that SREBP2 is either the sensor nor the only such mediator. Rather, SREBP2 is an important node that can be manipulated to inhibit the homeostatic response.

Action taken: We have added panel Figure 4E to the main text and two supplemental figures, in addition to extensive revisions of the text. See “Authors’ Response 3.2, 3.7, and 3.8” for details.

2- How were toxicity or differences in proliferation measured? (lines 141-142).

Authors’ response 1.2: The original manuscript relied on cell counting, without distinguishing between possible effects on proliferation versus toxicity. This weakness has now been extensively addressed. Please see “Authors’ Response 3.2 and 3.3” for details and actions taken.

3- I encourage the authors to show GP maps of cells treated with DHA at different time points after supplementation. It would be illustrative to compare with the control and let the readers appreciate the differences in the raw data.

Authors’ response 1.3: We agree. Representative images have been included in Fig 5 and Suppl Fig 15 for reference. Further, we have somewhat modified our analysis of the GP maps to include more pixels (by masking, rather than line scans) and thereby increasing signal/noise. This re-analysis (described in detail in the materials and methods) did not substantively affect the ultimate conclusions, as three different approaches yielded similar outcomes. See “Authors’ Response 2.7 and 2.8” below for more details.

Minor comments:

1- On the last paragraph of the Introduction (lines 91-100), it would be more informative if the authors add the key experimental approaches when summarizing the experimental findings that drove the conclusion. This would guide the reader since the beginning of the manuscript.

Authors’ response 1.4: We agree and have modified that paragraphs as follows.

Action taken: (pg 4) “Using shotgun mass spectrometry, we show that polyunsaturated fatty acids (PUFAs) are robustly incorporated into membrane phospholipids, introducing significant biophysical perturbations. This perturbation is counterbalanced by nearly concomitant lipidomic remodeling, most notable in the upregulation of saturated lipids and cholesterol. This remodeling normalizes membrane lipid packing, as evaluated by spectral imaging of the solvatochromic dye C-Laurdan.”

2- Authors need to format references on line 218.

Done

3- Figure 3 would be clearer if it included an inset (or scale break) highlighting the section between 0 and 18 hours to properly assess the simultaneous compensatory effect.

Authors’ response 1.5: Agreed, expanded panels showing the first 4 hrs of the responses are now included in Fig 3. Interestingly, closer analysis of this initial period suggests that the response we originally characterized as concomitant may actually be slightly delayed (~1 hr). We have thus changed the wording in that section to read “rapid remodeling” rather than “concomitant”. We thank the reviewer for their input.

Action taken: Edited Fig 3 and associated manuscript edits.

4- Legends should reveal what fits were used to adjust to the data.

Done

5- Rephrase “if this lipidome response were designed to...” (line 231), as it seems to point out to intelligent design.

Agreed! We have rephrased to “There, perturbations of membrane physical properties produced by changes in ambient temperature induce lipid changes whose apparent purpose is to re-normalize membrane physical properties.”

6- For clarity and reproducibility, include the whole western blots in figures 4D and S8.

Now included as new supplemental Figure S6.

Reviewer #2 comments:

In this study Levental et al. investigate homeostatic adaptation in mammalian cells, both at the cell and organ level using a combination of approaches – lipidomics, microscopy and Western blotting. They find that exposing cells to PUFAs results in their quick incorporation in membrane lipids, a change that within an hour is compensated by an increase in the level of saturated fatty acids and cholesterol as well as a decrease in other PUFAs than the ones supplemented. Similar changes in phospholipids are observed in mouse heart and liver. The transcription factor SREBP2 is implicated in the membrane lipid composition response in that its mature form is upregulated upon DHA-treatment, that the cholesterol increase is not seen in SRD-12B cells where it can not mature and after inhibition of SREBP2 processing by betulin. The study is mostly sound and the authors are correct in that not much is known about lipid remodeling in mammalian cells.

However, little of what is presented in this paper is new and the increased knowledge is incremental rather than substantial.

Authors' response 2.1: We thank the reviewer for their detailed analysis of our work. Although we agree that little is known about lipidomic remodeling in mammalian cells, we respectfully disagree that the presented results are either known or incremental.

First, we believe that the details and magnitudes of lipidomic perturbations induced by dietary fats are surprising and interesting. It is a rather remarkable fact that upon free-feeding of a fish oil-enriched diet, more than 30% of murine cardiac phospholipids contain a component that the mice cannot synthesize (i.e. an w-3 fatty acid; see Fig 1D). While changes induced by exogenous sources have been previously reported, the surprising extent of mammalian membrane susceptibility, especially *in vivo*, has only become accessible by the development of comprehensive lipidomics. Further, previous studies have tended to focus on remodeling of lipid acyl chains via fatty acid analysis¹⁻⁴, whereas our methods report effects on intact lipid species (allowing us to distinguish fully saturated lipids) as well as cholesterol concentration.

However, if this manuscript focused solely on lipidomics, we agree that it would not be appropriate for a general interest journal. The most novel and impactful aspects of our work are the observations of how cells respond to lipidomic perturbations and why they do so. Specifically, we report the first direct observations of homeostatic membrane adaptation in mammalian (or any other endothermic) cells. We directly experimentally confirm all major tenets of cell-autonomous biophysical membrane homeostasis: (1) lipidomic remodeling caused by a perturbation of membrane physical properties (Fig 2-4); (2) recovery of physical properties at a new lipid composition (Fig 5); and (3) necessity of this response for cellular fitness (Fig 6).

The fact that we observe similar responses across various mammalian cell types and also *in vivo* speaks to its general relevance. Thus, we believe our findings demonstrate a key and fundamental biological insight: mammals possess a cell-autonomous homeostatic response that maintains membrane physical properties upon perturbations from dietary lipids.

Finally, in this revision, we have added extensive experimental data to characterize the cytotoxic phenotype resulting from inhibition of membrane homeostasis (see Authors' Reply 3.2 and 3.3). We believe these data broaden our study into a robust and substantial contribution to the literature on this important topic.

1. There is no reason to speculate about the acyl chain composition in cell culture media. Using mass spectrometry lipidomics, it should be possible to estimate the levels of the supplemented fatty acids in the cell culture media and state the level of supplementation

Authors' response 2.2: We agree. Serum composition has been thoroughly documented⁵ and our speculation was correct: saturated and monounsaturated FAs are at least 10-fold more abundant in serum than long-chain polyunsaturated FAs like DHA or AA.

Action taken: Sentence edited to: “We expect that this disparity in incorporation between PUFAs and more saturated FAs is associated with their availability in cell culture media: cultured cells have access to sufficient levels of OA and PA such that supplementation at concentrations used here has no effect, whereas PUFA levels are limited such that supplementation with the physiologically appropriate concentrations used here leads to robust uptake and incorporation.”

2. In Figure 4 the mature form of SREBP2 gives a double band but in Figure S8 there is a single band for the same protein.

Please provide the entire Western blot to confirm that the correct bands are shown and explain the discrepancy. Moreover it is stated that there was no effect in the level of the precursor form of SREBP2, but it appears that the actin band (presumably the loading control) decreases in intensity at the longer times suggesting that there is a factual increase in the precursor form. Please quantify and discuss the results.

Authors' response 2.3: The graphs in Fig 4 D and E show the average of $n > 4 \pm$ SD. For each of these experiments, both the precursor and processed bands for SREBP1 or SREBP2 are normalized to the actin band within the same blot. Then the statistics for these normalized expression values are shown on the graph. Representative full-length Western blots are now shown in Supp Fig 6A-B and D-F.

We are unsure about the nature of the extra bands observed with the SREBP2 antibody, though they have been reported previously⁶⁻⁸. The bands at 55kDa and 68kDa both appear to represent mature processed forms of the protein. Quantifying either band gave very similar results, as shown in Supp Fig 6C.

3. Figure 5 raises a lot of questions: a) The effect of fatty acid supplementation on the GP-values in the plasma membrane are neither reported nor commented upon. Presumably because no change could be detected. Please report the results and discuss them.

Authors' response 2.4: The reviewer is correct. We did actually quantify the GP at the PM and observed rather minor changes (shown now as Fig 5E fix scale). The magnitudes of these changes were quite small and generally not statistically significant, thus we originally felt these results were peripheral to the main points of the paper. They have now been included and discussed in the main manuscript.

We have previously measured lipidomes of isolated PMs under these conditions. We observed similar lipidomic remodeling in isolated PMs as reported here for whole membrane lipidomes, including increased levels of saturated lipids and cholesterol. These are now included in Supp Fig 9.

Action taken: Added panel to Fig 5 showing GP changes induced by DHA in the PM and Supp Fig 9 showing lipidomic remodeling of isolated PMs. Added text describing these results and a potential explanation for why the PM is not as significantly affected as internal membranes: "A possible explanation for why PMs showed smaller changes than internal membranes is that exogenous PUFAs are preferentially incorporated into lipids at the ER, which is site of most lipid synthesis. If lipidome remodeling occurs faster than the trafficking of newly synthesized lipids to the PM, then biophysical disruptions of the PM may be minimal."

b) It is suggested in the main text, but not in the figure legend that panel C corresponds to 24h. Would it not then be the same as the last time point in panel B? However, the mean values and the spreads are not identical. Please explain why that is and state whether the GP-value in the +DHA and +betulin group is statistically different from the +DHA-group.

Authors' response 2.5: We apologize for the confusion. Panel 5C (now Panel 5F) shows the GP of whole cell homogenates using a spectrophotometer, as an independent confirmation of the GP imaging results.

Action taken: This information has now been clarified in the figure and legend. Also added text: "These spectral imaging observations were independently confirmed by cuvette spectroscopy of homogenized cell membranes labeled with C-Laurdan. As with imaging, neither 24 h treatments with DHA nor betulin alone affected membrane packing (GP) of whole cells, whereas cells treated with DHA in the presence of betulin had significantly reduced membrane packing (Fig. 5F)."

State whether the GP-value in the +DHA and +betulin group is statistically different from the +DHA-group.

Authors' response 2.6: All statistics are now shown in the figure panels and described in the Figure legends.

c) Please clarify what fraction of the total of the intracellular pixels were included in analysis after discarding all pixels with intensities lower than 3 standard deviations of the background value after the binning process. (Material and Methods page 21, lines 510-511).

Authors' response 2.7: We thank the review for this suggestion. Suppl Fig 15 now shows images representing each step of the process. 1. unbinned image, 2. binned image, 3. binned with background subtracted (5x5 pixel square defined by user), 4. binned with background subtracted and all pixels below 3SD of background thresholded to NA. As is evident from

these representative images, the signal-to-background is quite high, and very few pixels within a cell are lost after removing 3 standard deviations of background. This information has now been included in the Methods section.

Action taken: Added Supp Fig 15.

d) In panels A and B, please indicate what parts of the line scan was used for calculating the GP-values of the internal membranes.

Authors' response 2.8: We also thank the reviewer for this suggestion, which prompted us to re-address the quantification. To ensure that the specifics of our quantification method were not biasing the result, we quantified the GP of internal membranes by three different methods:

1. Line scans drawn across individual cells, as shown in the original figure. PM GP values were taken as peak GP values from the periphery of the cell, whereas internal membranes were calculated as the average of all values inside the PM peaks.
2. The nucleus is visible as a dark spot in C-Laurdan images. We used this as a fiducial marker to draw a region-of-interest representing peri-nuclear membranes and calculated the GP of those pixels.
3. As shown in the images in the updated Figure 5, the internal membranes were manually masked (dotted lines in Fig 5A), and histograms of the pixels inside were plotted.

Ultimately, all three methods produced quantitatively similar results, suggesting that this measurement is robust. We have chosen the third method because it includes the most information, allowing us to evaluate the distributions of GP values.

Finally, we also independently measured the C-Laurdan GP of total cellular membranes using spectroscopy of homogenized cells (Fig 5F). This information is now included in the figure legend and text.

Action taken: Figure 5, and the accompanying legend, text, and Methods section, have been significantly modified from the previously submitted version. We have added the above discussion on various ways of quantifying C-laurdan GP to the Methods section.

4. For Figures S4 and S5 statistics need to be supplied in order to allow any statements on decreases in gene expression and lipid saturation.

Authors' response: Additional experiments were performed and these two Supplementary Figures (and all others) were updated with statistics.

5. The data in Figure S6 suggests that betulin treatment increases the fraction of saturated lipids. Please provide the statistics and comment upon the results.

Authors' response: This result was from N=2 and did not hold up upon a further two repeats. Thus, we have removed it from the manuscript.

Minor points

1. In Figure 6, the y-axes of the A- and B-panels are labeled incorrectly.

Authors' response: This figure has been completely changed and these issues were addressed.

2. Figure S1. Please clarify if the data are new or copies from the 2017 paper.

Authors' response: The data for MSCs were published in a previous manuscript; all others are new. This clarification was added to the figure legend.

3. In Figure S2 panels A and B the labelling of +PA and +OA does not match the figure legend.

Authors' response: Thank you for the careful reading; this was fixed.

4. In Figure S7, something odd has happened with the last data point and there appears to be both a mix-up of the symbols and a missing data point.

Authors' response: Fixed.

5. Please comment upon why SRD12B cells by default were maintained in media supplemented with mevalonate, oleic acid and cholesterol. (Page 18 lines 407-408).

Authors' response: Due to their inability to process SREBP, SRD12B cells are auxotrophic for cholesterol, mevalonate, and unsaturated fatty acids. These must be included in the media for normal cell growth: from Rawson et al⁹, where this cell line was first described, “the mutant cells are rescued by the addition of exogenous cholesterol, mevalonate, and unsaturated fatty acids, under which conditions they grow with nearly normal kinetics.”

Action taken: A note has been added to the Methods.

6. Please explain why the diet contained glycerol and propylene glycol. (Page 18, lines 430-431).

Authors' response: The rodent diets included the food-grade antioxidant Tenox20A at 0.1% w/w to guard against fatty acid oxidation. These components are part of the Tenox 20A formulation.

Action taken: To avoid confusion for readers, the detailed description of ingredients in Tenox 20A was removed.

7. Line 349 should read temperature rather than temperatures. Line 465 should read 4°C rather than 4_C

Authors' response: Thank you, fixed.

Reviewer #3 (Remarks to the Author):

“Homeoviscous adaptation” is the preservation of membrane properties despite a changing external environment. Changes in membrane lipid composition in response to temperature fluctuations are well characterized in ectothermic species, but processes governing mammalian membrane homeostasis are less studied, as mammals maintain a somewhat constant internal body temperature. Here the authors provide experimental evidence showing that mammals actively modulate their lipidomes in response to changes in dietary lipid. In general, highly unsaturated omega-3 and omega-6 fatty acids from the diet are preferentially incorporated into membrane lipids, and the authors demonstrate that cells respond by increasing incorporation of cholesterol and unsaturated lipids into membrane lipids as well. The authors claim that this is the first demonstration of homeoviscous adaptation in response to dietary lipids in mammals, and thus has important implications for the understanding of mammalian membrane homeostasis. The authors seek to demonstrate that (i) dietary PUFAs are preferentially incorporated into membrane lipids, (ii) in response, cells actively incorporate stabilizing lipids into their membranes, and (iii) these changes are essential for cellular fitness. The first point is well supported in the existing literature, so these experiments are important controls but not novel findings. The second point is both novel and well-supported by the data, showing that lipidomes are remodeled following PUFA incorporation...

Authors' response: We thank the reviewer for their careful analysis of our work and positive feedback. We would only point out that biophysical membrane perturbations induced by dietary fatty acids, and their re-normalization by lipidomic remodeling, were an important and novel aspect of the manuscript. These have been expanded upon in the revision via quantification of membrane permeability in live cells.

... mechanistic insight is relatively shallow (it is shown to be SREBP-dependent)... cell growth in culture was the only metric of cellular fitness reported and is therefore not sufficient to be broadly indicative of cellular fitness. As it is written, the claims and conclusions of the paper exceed what is supported by the data and appear more suited for a more specialized journal

Authors' response 3.1: We thank the reviewer for their critical comments, which have encouraged us to address these weaknesses and thereby significantly improve our manuscript. These changes have focused on three distinct areas:

1. In depth analysis of the “cell fitness” phenotype, identifying non-apoptotic cell death as the underlying cause
2. Development of a novel assay to quantify the effect of lipid perturbations on membrane permeability
3. WB and qPCR analysis of additional mediators that could be involved in the homeostatic response

The results of these investigations are described in detail below. All have sharpened the original conclusions and added depth to the manuscript.

In the title, as well as throughout the text, the authors state that lipidome remodeling is essential for “cellular fitness”. However, the only assay used to evaluate cellular fitness is growth rate (relative cell number after 3 days, Fig. 6C). In order to make claims about cellular fitness, the authors would need to demonstrate: a. Genetic or cytological indicators of reduced cellular fitness or increased cellular stress (independent of growth rate).

Authors' response 3.2: We fully agree and have performed extensive new experiments to determine the specific nature of the reduced “cellular fitness”.

Cell cycle: first, we examined whether DHA and/or betulin had effects on cell cycle using a flow cytometric assay to quantify the relative DNA abundance in individual cells. As shown in Supplemental Figure S11, neither DHA, betulin, nor their combination had any significant effect on cell cycle progression. These results suggested that the reduced fitness was not the result of slowed cell cycle progression, but rather cytotoxicity.

Cell viability: The above inference was confirmed by cell counting and trypan blue (TB) staining. In RBL cells, neither DHA nor betulin alone affected the number of viable cells (Fig. 6A) nor significantly increased the number of TB+ (i.e. dead) cells (Fig. 6B). In contrast, the combined treatments significantly reduced the number of viable cells and significantly increased the TB+ cells. These observations were mirrored in SRD12B cells, which cannot mobilize SREBP responses (Fig. 6C-D) in that DHA induced significant reductions in cell numbers and increased TB+ cells. Thus, perturbation of membrane properties combined with inhibition of the homeostatic response results in significant cytotoxicity.

Induction of Unfolded Protein Response (UPR): we hypothesized that uncompensated membrane perturbations may evoke sustained UPR and associated apoptosis. Using qPCR, we evaluated DHA/betulin induced changes in ATF6 target genes (HSP90B and GRP78) and IRE1-mediated XBP1 splicing. These represent distinct UPR pathways, with the ATF6 pathway particularly associated with UPR-induced apoptosis. We observed no significant changes in any of these readouts, suggesting that UPR is not involved in either DHA-induced remodeling or the cell fitness phenotype. This data is now shown in Fig S8.

Apoptosis: DHA+betulin increased the percentage of PI-positive cells (6C), consistent with the cell death described above. One hallmark of apoptosis is the externalization of PS (marked by Annexin V) without loss of membrane integrity (marked by PI), allowing apoptotic cells to be identified as AnxV+ / PI- by flow cytometry. After 24 h of treatment with DHA and betulin alone or in combination, we observed no increase in AnxV+ / PI- cell numbers (Fig 6D-G), suggesting that the increased cell death was not apoptotic. Consistently, using a fluorescent marker of active caspases (CellEvent™ Caspase-3/7 Green fluorescence), we saw no induction of apoptosis in RBL cells treated with DHA+betulin (Suppl Fig 12A). Finally, the involvement of apoptosis in a cytotoxic process can be inferred by using potent caspase inhibitors (e.g. Z-VAD-FMK), which should attenuate apoptosis-induced cytotoxicity. Treatment with this inhibitor did not inhibit cell death (Fig S12B), supporting a non-apoptotic cytotoxicity mechanism.

Together, these results reveal that inhibiting lipidomic remodeling in the presence of DHA results in non-apoptotic cell death, presumably through a necrotic effect.

b. Extend these findings to demonstrate cytotoxicity and/or show some kind of physiological response in vivo when remodeling is blocked

Authors' response 3.3: The effects on cytotoxicity are discussed above. We feel that in vivo experiments are beyond the scope of this work. However, our observations also prompted us to explore the possible causes of non-apoptotic cell death induced by uncompensated membrane perturbations. Although biophysical membrane perturbations likely have several distinct physiological effects, we considered the central role of the membrane as a selective barrier. Significant computational and model membrane literature^{10,11} suggests that membrane permeability is highly affected by lipid composition, with more loosely packed, fluid membranes being more permeable to various polar and amphiphilic compounds. Simulations have specifically analyzed polyunsaturated phospholipids and shown that DHA-containing lipids are 2-3-fold more permeable than monounsaturated counterparts¹⁰. Thus, we hypothesized that the permeability of cell membranes would be reduced by incorporation of polyunsaturated lipids. To test this possibility, we developed a method to quantitatively evaluate changes in membrane permeability to amphiphilic substrates in living cells.

Specifically, we modified a classical cell viability assay that relies on the passive diffusion of the amphiphilic compound fluorescein diacetate (FDA) through the PM into the cytoplasm, where cellular hydrolases convert it into fluorescent fluorescein whose charge prevents it from diffusing out of the cell¹². Because the catalytic conversion of FDA is highly efficient¹², the rate of fluorescence increase is a direct readout of FDA flux through the PM. We observed a linear increase in fluorescence over at least the first 10 mins of measurement, consistent with a constant flux (Q). The permeability coefficient (P) of a membrane to a particular substrate is then defined by Fick's Law as: $Q = P * A * (C_{out} - C_{in})$, where A is the area of the membrane and C_{out} and C_{in} are the concentrations of the solute outside and inside the cells, respectively. In our experiments, $C_{in} = 0$ because FDA is quickly converted into a different molecule upon entering the cell.

We estimate the transport surface area by multiplying a typical mammalian surface area ($\sim 3000 \mu\text{m}^2$) by the number of cells in the cuvette (2.5×10^5). Using pure fluorescein for calibration, we measured directly the flux of FDA molecules across the membrane, which for typical sample of untreated RBL cells was 6.6×10^{-5} nmol/sec. From these parameters, we calculated a permeability coefficient for FDA to be $\sim 2.2 \times 10^{-6}$ cm/sec, in excellent agreement with values previously reported in plant cells¹³. We found that this flux scales linearly with cell and FDA concentrations, consistent with the simple Fick's Law model of passive diffusion followed by rapid active conversion.

Finally, we tested the effects of DHA or betulin alone, or both treatments combined to perturb the membrane while inhibiting the homeostatic response. Consistent with our hypothesis, we observed that cells with uncompensated membrane perturbations are indeed more permeable (new Fig 5H-I). Because cell physiology is essentially dependent on maintaining cytoplasmic concentrations of a plethora of small molecules (water, alcohols, polyamines) and amphiphiles (steroids, fatty acids) that can passively diffuse through the membrane, it is probable that the significant increases in membrane permeability may be responsible for the observed necrosis.

Action taken: The above discussion and associated Figures have been added to the manuscript.

c. Eliminate the possibility that DHA/lipidome remodeling directly modulates rates of cell division (e.g. via a mitogenic pathway, rather than indirectly through changes in cellular fitness). Direct modulation of division rates seems particularly probable given that low-dose DHA appears to stimulate cell division in WT cells (see fig. 6C), contrary to statements made in the results section (line 141-142, stating that no "obvious ... differences in proliferation" were observed).

Authors' response 3.4: The effects on cell cycle progression were directly quantified and no significant effects of any of the treatments were observed, as described in Sec 3.2 above.

Regarding the seeming contradiction between Fig 6C and the text, we apologize for the confusion. Lines 141-142 referred to RBL cells, in which we observed no notable differences in proliferation with DHA (now Fig 6A and Suppl Fig S3) at low concentrations, and DHA being somewhat cytostatic at higher concentrations.

The graph in 6C (current manuscript) compares WT CHO cells to the SRD12B mutant cells supplemented with DHA. In those experiments, both lines were cultured in SRD12B medium (i.e. supplemented with mevalonate, oleic acid, and cholesterol) to eliminate media conditions as a source of variation. Under these conditions, DHA does slightly increase the proliferation of the WT CHO cells at low concentrations. The reasons for this are beyond the scope of this study.

2. Data are not quantified correctly. Figures 1 A, B, D, and E have Y-axes that measure a particular fatty acid based on chain length and degree of unsaturation (22:6 for DHA supplementation experiments, and 20:4 for AA supplementation). However, Supplemental figure 3A shows that both AA and DHA are desaturated significantly prior to incorporation into glycerophospholipids (for example $\sim 50\%$ of the DHA appears to be incorporated as 22:6, but the remaining 50% of incorporated DHA appears to be in the form of fatty acids with >6 degrees of unsaturation). Therefore, it may be more appropriate to quantify changes in omega-3 and omega-6 fatty acids, rather than changes in 22:6 and 20:4 fatty acids. Ideally, radiolabeled fatty acids could be used to directly track the incorporation of fatty acids into different species even after remodeling.

Authors' response 3.5: We apologize for the confusion caused by our reporting of the lipidome. While the MS/MS lipidomics identifies both acyl chains of a lipid independently, we typically report total lipid unsaturation, i.e. the combined number of double bonds in both acyl chains of a given lipid specie. Thus, lipids with 6 unsaturations are usually those where DHA is combined with a saturated FA partner (e.g. palmitate) while those with >6 unsaturations are essentially exclusively DHA and a mono- or diunsaturated FA partner. Mammalian cells cannot further desaturate DHA, and we did not observe any lipids containing acyl chains with more than 6 unsaturations.

However, the reviewer is correct that some FA remodeling can occur, usually retroconversion of DHA into shorter w-3 PUFAs like EPA (20:5). We have quantitatively evaluated this effect and found it to be relatively minor, i.e. $<5\%$ of supplemented DHA is incorporated as a remodeled FA. While we agree that it would be best to evaluate all possible w-3/w-6 species, MS/MS does not provide the location of the double bonds, only their number and chain length. Thus, we focus on those species that we can unambiguously identify, i.e. DHA, EPA, and AA.

3. The discussion of in vivo experiments is misleading, and the experimental design is questionable. Interpretation of the in

in vitro studies leads readers to understand that dietary omega-3 and omega-6 PUFAs are “robustly and specifically incorporated ... into membrane lipids” (section 1 of the results), which leads to a concomitant increase in unsaturated lipids to maintain membrane homeostasis. Then, *in vivo* experiments compare fish oil supplementation to corn oil supplementation. In the main text, corn oil is described as a poor source of omega-3 PUFAs (line 122). However, corn oil is a very rich source of omega-6 PUFAs in the form of linoleic acid (18:2). If readers were told that corn oil is a rich source of Omega-6 PUFAs, they might expect it to induce a similar lipidome remodeling as seen in fish oil, but strangely such an effect is not observed (Fig. 2D). This is an interesting result, which seems to indicate that highly unsaturated omega-3/6 fatty acids (DHA and AA) induce lipidome remodeling, whereas the more saturated omega-6 PUFA (LA) does not.

The main text needs to be modified to highlight the omega-6 PUFA content of corn oil, and the resulting discussion needs to be significantly restructured to emphasize differences between moderately and highly unsaturated PUFAs.

Additional experiments may need to be performed if the authors wish to demonstrate differences in lipidome remodeling in response to level of unsaturation.

Authors’ response 3.6: We thank the reviewer for pointing this out. This point is correct; our descriptions of the *in vivo* feeding experiments were unintentionally mis-leading by focusing on w-3 PUFAs in the fish oil. However, we believe that the experimental design is appropriate to highlight the important result that the reviewer identifies: abundant dietary long-chain PUFAs cause robust lipidomic remodeling consistent with observations *in vitro*. Corn oil is indeed rich in linoleic acid, but poor in long-chain, highly unsaturated PUFAs: only 1.3% of CO fatty acids bear 3 or more unsaturations, compared to 22.5% in FO (Table S1). Consistently, FO feeding led to accumulation of lipids with highly unsaturated fatty acids (acyl chains containing 5 or 6 double bonds; new Fig 1E) and an associated increase in unsaturation index (Fig 1F). The robust remodeling we observe with FO feeding suggests that it is indeed these highly unsaturated lipids which affect membrane properties and induce the compensatory response. This inference is consistent with previous biophysical measurements suggesting that the extent of unsaturation scales with effects on membrane physical properties¹⁴.

The optimal set of experiments would control dietary lipid inputs across a broad set of conditions, including complete exclusion of PUFAs. These experiments will be the basis of our future work.

Action taken: We have modified Figure 1, the associated legend, and the text.

Pg5: “To confirm that the membrane incorporation of supplemented PUFAs in cultured cells appropriately recapitulates *in vivo* conditions, we analyzed membrane lipidomes in mouse cardiac tissue after two weeks of *ad libitum*-feeding on a semi-purified diet containing either fish oil (FO) or corn oil (CO), as previously described². The FO diet is highly enriched in long-chain w-3 PUFAs, with DHA and eicosapentaenoic acid (EPA; 20:5) comprising ~20% of all fatty acids. In contrast, CO is poor in w-3 PUFAs and long-chain PUFAs in general (1.3% FAs with 3+ unsaturations), with w-6 linoleic acid (18:2) being the major PUFA source (>50% of dietary FAs) (see Supplement Table 1 for full dietary lipid profile). Consistent with previous observations^{2,4,15} and the *in vitro* measurements here, dietary fish oil supplementation produced a robust incorporation of w-3 PUFAs into cardiac membrane lipids, with ~18-fold more w-3 PUFA-containing membrane lipids in the FO-fed compared to CO-fed tissues (Fig 1D). As a result, FO-fed mice had a significantly higher proportion of lipids containing very highly unsaturated acyl chains (5 or 6 double bonds; Fig 1E), which resulted in an overall increase in the unsaturation index in FO-fed tissues (Fig 1F).

Pg6: “Importantly, this response is also evident *in vivo*. The incorporation of highly polyunsaturated PUFAs from the diet into membrane lipids in mouse cardiac tissue (Fig 1D-E) led to remarkably similar lipidomic remodeling as was observed in cultured cells. Namely, we observed that FO fed-mice have ~3-fold more saturated lipids compared to CO-fed mice, fully consistent with compensatory, homeostatic lipidomic remodeling *in vivo*. The striking differences between FO and CO feeding are notable in light of the fact that CO diets do contain abundant PUFAs in the form of linoleic acid (18:2). These differences are potentially due to the unique effects of highly polyunsaturated lipids on membrane properties, which scale with the degree of unsaturation^{14,16}. An alternative, non-exclusive possibility is that highly unsaturated FAs are preferentially taken up into membrane lipids. These possibilities will be resolved by future studies with various FA feeding protocols.”

4. *SREBP* experiments are incomplete. Only the *SREBP2* isoform was studied (fig. 4C), however the experimental manipulations performed (*betulin* and *SRD12B*) interfere with *SREBP1(a and c)* as well. Moreover, *SRD12b* disrupts all transcription factors that are golgi processed by *SIP* (e.g. *CREBH*, *ATF6*...) not just *SREBP*. This is important because these other transcription factors have known effects on ER biology, lipid metabolism and trafficking (see Kim et al. *Mol Cell Biol.* 2017 Jun 29;37(14) and Cheng et al. *J Biol Chem.* 2016 Nov 4;291(45):23793-23803). Clarifying this point is particularly relevant given that disruption of *SREBP* processing causes a “failure to upregulate saturated lipids” (line 364), which is “not directly connected to *SREBP2* target genes”, but is directly connected to *SREBP-1c* target genes.

Authors' response 3.7: We agree. SREBP2 was originally chosen for analysis because of the DHA-induced upregulation of cellular cholesterol. In the previous version of the manuscript, genetic and pharmacological inhibition of SREBP processing abrogated the lipidomic response, leading us to conclude that SREBP2 was an important node in the homeostatic membrane mechanism. However, we did not intend to imply that SREBP2 is either a sensor, a sole regulator, or even the most important mediator of this response. The membrane remodeling we observe is comprehensive, involving the majority of the lipidome and suggesting many layers of control, likely including transcriptional, enzymatic, and trafficking changes. As the molecular mediators of the homeostatic response were not a central focus of the original manuscript, the observation that SREBP2 was essential allowed us to instead focus on the cellular effects of abrogation of membrane homeostasis. We did not make this point clearly in the original manuscript, rather placing too much emphasis on the role of SREBP2, which we believe to be important (possibly essential) but certainly not unique or sufficient. This emphasis has been addressed in the revision.

However, we agree with the reviewer that this aspect of the manuscript was incomplete. As discussed above in Authors' response 1.1 and 3.2, we further analyzed the effects of DHA on SREBP1, ATF6, and other UPR readouts and found no significant effect of DHA on any of these outputs. We have added these results to Figures 4, S6, S8, S10, and S12. We have also discussed our observations in the context of previous publications on the distinct effects of PUFAs on SREBP1 versus SREBP2 regulation.

Action taken: Amendment of Fig 4 and addition of three Supplementary Figures.

Added to Results: "Having observed a robust and rapid increase in cholesterol levels resulting from DHA supplementation, we evaluated whether SREBP2 processing was associated with this response. Indeed, DHA feeding increased the production of the 'mature' transcription factor form of SREBP2, with minimal effect on the precursor form (Fig 4D). In contrast, the proteolytic processing of SREBP1, the SREBP family member primarily implicated in unsaturated fatty acid metabolism¹⁷, was not affected by DHA under our conditions. The relationship between these observations and previous reports of PUFA-regulation of SREBPs are expanded upon in the Discussion.

... It is important to point out that S1P also processes other important signaling proteins like ATF6, responses that would also be abrogated in SRD-12B cells (though likely not by betulin treatment). To evaluate whether ATF6 may also be associated with DHA-induced responses, we tested the effect of DHA on proteolytic processing of ATF6 and observed no induction (Fig S8). We also observed no significant upregulation of the ATF6 target gene Grp78. Finally, we did not observe induction of HSP90B (ER stress response gene) or processing of XBP1, suggesting that ER stress is not significantly induced by DHA feeding (Fig S?). While these results support a role for SREBP2 processing in PUFA-induced lipidome remodeling, they definitively do not rule out other contributors to the homeostatic response. Nor do they implicate SREBP2 as the sensor for membrane fluidity. Rather, they establish SREBP2 as a node of the homeostatic response, and one that can be manipulated to evaluate manipulations of membrane homeostasis."

Moreover, DHA supplementation has been frequently associated with suppression of SREBP1 and reduction in expression of desaturase genes (e.g. see Haung et al., BMC Cancer. 2017; 17: 890 and Deng et. al Biochim Biophys Acta. 2015 Dec;1851(12):1521-9 and Caputo J Cell Physiol. 2011 Jan;226(1):187-93). In sum, simply showing that SREBPs is involved is not enough given this extensive literature. Especially since the authors are arguing that DHA's upregulation of membrane cholesterol by SREBP2 is the key driver of the adaptation.

Authors' response 3.8: The stimulation of SREBP processing and gene regulation by PUFA-induced membrane remodeling was a somewhat surprising result, as DHA has been previously shown by others to suppress SREBP1 activation, while having minimal effects on SREBP2. It is still unclear in the literature how these two family members are differentially regulated. The previous version of this manuscript was somewhat vague in emphasizing these differences, rather treating both as "SREBPs". This has now been corrected.

The seeming discrepancy between our induction of SREBP2 processing and previous reports of no notable effects of DHA on SREBP2 is likely explained by differences in experimental conditions: in previous reports, experiments were performed under acute PUFA stimulation of serum-starved cells. In these conditions, high levels of activated SREBP2 induced by lack of exogenous cholesterol likely suppress any further stimulation by DHA. We have confirmed that in our cells under low serum conditions, cleaved SREBP2 is high and DHA does not induce any further proteolytic processing (now shown as Fig S14). The experiments described in our manuscript were done by supplementing full serum with physiologically relevant PUFA concentrations. Under these conditions, basal SREBP2 processing is low and is significantly induced by DHA

supplementation (see Fig 4). These results support a role for SREBP2 in homeostatic lipidomic remodeling and resolve the seeming discrepancy with previous observations.

Action taken: Added supplemental figure and Discussion: “The effects of PUFA supplementation on SREBP signaling have been extensively studied, and DHA is known to suppress the activation of SREBP1 and its target genes, both in cultured cells and *in vivo*^{6,18,19}. In contrast, SREBP2 is not suppressed by PUFAs^{6,18}, revealing that these two complementary regulators of membrane homeostasis have different functions, despite both being sensitive to membrane cholesterol. In our observations, SREBP2 is induced as an important part of the homeostatic response. Previous reports⁶ have not noted a significant effect of PUFAs on SREBP2, possibly because those experiments involved acute PUFA feeding of serum-starved cells, where high levels of activated SREBPs may have suppressed the DHA-mediated stimulation we observe. We have confirmed this effect in our cells (Fig S14).”

5. There is a rich literature on PUFA supplementation in both cultured cells and in whole animals. Very few of these studies report the kind of changes in saturated fatty acids reported by the authors. For example, Harris et al report fatty acid data from people with heart biopsies following 6 months of w3 PUFA supplementation where they found no change in saturated fatty acids, an increase in w3 FAs (as expected due to the supplementation) and a reduction in w6 FAs (Circulation. 2004;110:1645–1649). Similarly, Owen et al., reported on rats fed from 0-42days a high w3 diet really did not see significant changes in myocardial saturated fatty acids. Yet, they did observe a major increase in w3 FAs and an associated reduction in w6 FAs (Lipids , 2004 Vol. 39, Iss. 10). In short, the authors discuss little of this vast literature of PUFA supplementation and how it does or does not agree with their findings.

Authors’ response 3.9: We thank the reviewer for pointing this out, as we had also previously analyzed published studies on PUFA supplementation and were surprised that this effect had not been noted. As it happens, in many of these studies, saturated FAs are indeed increased while other unsaturated FAs are decreased. Even in the two studies cited by the reviewer:

- Harris et al (cardiac tissue in humans) found stearic acid (C18:0) increased by >10% with w-3 PUFA supplementation, with similar reductions in oleic and linoleic acids. This was despite a much smaller incorporation of DHA (probably due to smaller supplementation dose).
- Owen et al (cardiac tissue in rats) found that saturated fatty acids as a class are increased by ~20% upon fish oil feeding, whereas MUFAs are reduced by ~50%, as are w-6 PUFAs. Similar effects, with slightly smaller magnitudes, are reported for erythrocyte lipids.
- Broadly similar effects are also evident across a number of other publications¹⁻⁴.

There are several reasons why this effect may have been previously overlooked:

1. The remodeling effects (e.g. upregulated of saturated lipids) are obviously less pronounced than the “main” effects (e.g. increased DHA and decreased AA) and so may have been considered incidental/irrelevant
2. The large majority of previous reports have relied on FAs isolated from lipids by hydrolysis. Analyzing isolated FAs rather than intact lipids can obscure membrane remodeling because storage lipids and/or free fatty acids do not contribute to membrane properties but can comprise a large fraction of fatty acids in a tissue sample. Moreover, isolated FAs do not reveal how they are combined into phospholipids, so it is possible to get more fully saturated lipids without a major increase in saturated FAs (though this is not what we observed in our samples)
3. Remodeling of the non-supplemented FAs can also be obscured by the high abundance of the supplemented FAs. For example, if >30% of the phospholipidome is converted into w-3 PUFA containing lipids by DHA feeding, then the mol fraction of other FAs must obviously decrease. Most of the w-3 PUFAs are replacing w-6 PUFAs, but not all. If all highly polyunsaturated PUFAs are removed from the analysis, then the increase in SFAs by DHA feeding in previous studies becomes more obvious.
4. Cholesterol is rarely considered alongside FAs and is a major component of the remodeling we report.

Action taken: added to Discussion: “With extensive literature on the effects of dietary lipids on membrane compositions^{1-4,20}, it is somewhat surprising that the remodeling we observe has not been previously reported. We note that in many of the published datasets, increases in saturated fatty acids have indeed been observed, though rarely remarked upon. We speculate that the compensatory effects may have been previously overlooked for several reasons: (a) remodeling effects are obviously less pronounced than the “main” effects (incorporation and substitution) and may have been considered incidental/irrelevant; (b) most previous reports have analyzed FAs rather than intact lipids. Isolated FA analysis could obscure membrane remodeling because storage lipids and/or free fatty acids do not contribute to membrane properties but can comprise a large fraction of fatty acids in a tissue sample; (c) even for isolated membrane lipids, FA analysis does not reveal how these are combined into phospholipids, so it is possible to get more fully saturated lipids without a major increase in saturated FAs; (d) remodeling of the non-supplemented FAs can be obscured by the high abundance of the supplemented FAs. For example, if a large fraction of the phospholipidome is converted into DHA-containing lipids by DHA feeding, then

the mol fraction of other lipids obviously decreases, even if the relative proportion of saturated acyl chains compared to unsaturated is increased; (e) cholesterol is rarely considered alongside FAs and is a major component of the remodeling we report.”

6. *Statistics are not reported for the majority of the supplemental figures, as well as several figures in the main text (Fig. 3, Fig. 4 C-D).*

Action taken: We have added statistical analyses to all appropriate figure panels.

Minor:

1. *Supplemental figure 3A shows a significant decrease in monounsaturated lipids in response to AA treatment. It is unclear why this result is not mirrored in the corresponding figure in the main text (Fig. 2a), where monounsaturated lipids appear unchanged. This discrepancy needs to be discussed in the main text.*

Authors’ response 3.10: We apologize for the confusion. Supp Fig S3A (now Supp Fig S5) shows the distributions in “raw” lipidomes, whereas the original Figs 2A-B showed the distribution of only those lipids remaining after lipids containing the supplemented FAs (DHA or AA) were removed from the analysis. We believed this “remaining lipids” analysis was most appropriate for specifically highlighting the remodeling induced by supplemented FAs. However, we have realized that this analysis is confusing to almost everyone (including ourselves) and have thus removed it. We now show the raw lipidome data as a mol% of all GPLs.

2. *Lines 163 and 164 state that there was “a notable increase in relatively short non-DHA/AA containing lipids (Fig. S3D-E)”. However, no statistical test was described or reported to support this conclusion. It is therefore unclear whether there is a significant difference (in which case statistics need to be reported), or whether the authors wish to highlight a non-significant trend (which needs to be clearly qualified in the main text).*

Action taken: This increase was only evident when the supplemented FAs were removed from the analysis. In the raw lipidomes, the effects were not significant and not notable enough to be mentioned. Thus, these data were removed.

3. *The authors claim that PUFAs were not incorporated into TAGs, but this data is not shown. These data provide essential context for our interpretation of the degree of incorporation into GPLs.*

Authors’ response 3.11: We thank the reviewer for pointing this out. The original statement was referring to cell culture data, where storage lipids are generally minimal and almost never contain very highly polyunsaturated tails. Our lipidomics data does not directly define the molecular composition of the three FA chains in TAGs and therefore cannot say which FAs are present, rather only infer from the intact molecule whether PUFAs are present (e.g. TAG 54:2 cannot have AA or DHA). The tissue samples have more storage lipids, which generally have higher more unsaturations, so we cannot confidently claim anything about their actual FA composition. Because of this, and because this remark was tangential to our manuscript, we have removed it.

4. *There is insufficient discussion of membrane compartments. It is unclear why the GP is reported only for internal membranes (Fig. 5A), as it would be an interesting point of discussion if the plasma membrane composition responded differently than internal membranes.*

Action taken: We agree and have added data and discussion on the lipidomic remodeling and biophysical changes in the PM. Please see “Authors’ Response 2.4”.

References

- 1 Fan, Y. Y. *et al.* Dietary docosahexaenoic acid suppresses T cell protein kinase C theta lipid raft recruitment and IL-2 production. *J Immunol* **173**, 6151-6160, (2004).
- 2 Fan, Y. Y., McMurray, D. N., Ly, L. H. & Chapkin, R. S. Dietary (n-3) polyunsaturated fatty acids remodel mouse T-cell lipid rafts. *The Journal of nutrition* **133**, 1913-1920, (2003).
- 3 Atkinson, T. G., Barker, H. J. & Meckling-Gill, K. A. Incorporation of long-chain n-3 fatty acids in tissues and enhanced bone marrow cellularity with docosahexaenoic acid feeding in post-weanling Fischer 344 rats. *Lipids* **32**, 293-302, (1997).
- 4 Stark, K. D. *et al.* Fatty acid compositions of serum phospholipids of postmenopausal women: a comparison between Greenland Inuit and Canadians before and after supplementation with fish oil. *Nutrition* **18**, 627-630, (2002).
- 5 Psychogios, N. *et al.* The human serum metabolome. *PloS one* **6**, e16957, (2011).
- 6 Hannah, V. C. *et al.* Unsaturated fatty acids down-regulate srebp isoforms 1a and 1c by two mechanisms in HEK-293 cells. *J Biol Chem* **276**, 4365-4372, (2001).
- 7 Sheng, Z., Otani, H., Brown, M. S. & Goldstein, J. L. Independent regulation of sterol regulatory element-binding proteins 1 and 2 in hamster liver. *Proc Natl Acad Sci U S A* **92**, 935-938, (1995).
- 8 Ye, J. *et al.* Asparagine-proline sequence within membrane-spanning segment of SREBP triggers intramembrane cleavage by site-2 protease. *Proc Natl Acad Sci U S A* **97**, 5123-5128, (2000).
- 9 Rawson, R. B., DeBose-Boyd, R., Goldstein, J. L. & Brown, M. S. Failure to cleave sterol regulatory element-binding proteins (SREBPs) causes cholesterol auxotrophy in Chinese hamster ovary cells with genetic absence of SREBP cleavage-activating protein. *Journal of Biological Chemistry* **274**, 28549-28556, (1999).
- 10 Venable, R. M., Kramer, A. & Pastor, R. W. Molecular Dynamics Simulations of Membrane Permeability. *Chem Rev* **119**, 5954-5997, (2019).
- 11 Manni, M. M. *et al.* Acyl chain asymmetry and polyunsaturation of brain phospholipids facilitate membrane vesiculation without leakage. *eLife* **7**, (2018).
- 12 Rotman, B. & Papermaster, B. W. Membrane properties of living mammalian cells as studied by enzymatic hydrolysis of fluorogenic esters. *Proc Natl Acad Sci U S A* **55**, 134-141, (1966).
- 13 Baron-Epel, O. *et al.* Dynamic continuity of cytoplasmic and membrane compartments between plant cells. *J Cell Biol* **106**, 715-721, (1988).
- 14 Lin, X. *et al.* Domain stability in biomimetic membranes driven by lipid polyunsaturation. *J Phys Chem B* **120**, 11930-11941, (2016).
- 15 Metcalf, R. G. *et al.* Effects of fish-oil supplementation on myocardial fatty acids in humans. *The American journal of clinical nutrition* **85**, 1222-1228, (2007).
- 16 Filippov, A., Oradd, G. & Lindblom, G. Domain formation in model membranes studied by pulsed-field gradient-NMR: the role of lipid polyunsaturation. *Biophys J* **93**, 3182-3190, (2007).
- 17 Raghow, R. *et al.* SREBPs: the crossroads of physiological and pathological lipid homeostasis. *Trends Endocrinol Metab* **19**, 65-73, (2008).
- 18 Yahagi, N. *et al.* A crucial role of sterol regulatory element-binding protein-1 in the regulation of lipogenic gene expression by polyunsaturated fatty acids. *J Biol Chem* **274**, 35840-35844, (1999).
- 19 Xu, J. *et al.* Polyunsaturated fatty acids suppress hepatic sterol regulatory element-binding protein-1 expression by accelerating transcript decay. *J Biol Chem* **276**, 9800-9807, (2001).
- 20 Owen, A. J., Peter-Przyborowska, B. A., Hoy, A. J. & McLennan, P. L. Dietary fish oil dose- and time-response effects on cardiac phospholipid fatty acid composition. *Lipids* **39**, 955-961, (2004).

Reviewers' Comments:

Reviewer #1:

Remarks to the Author:

The revision submitted by Levental et al., in my opinion, properly answers the reviewers comments. The manuscript "Lipidomic and biophysical homeostasis of mammalian membranes in response to dietary lipids is essential for cellular fitness" is very solid and timely and it will have a great influence in the field.

Reviewer #2:

Remarks to the Author:

Summary

In the revised version of their manuscript Levental et al. present more experimental evidence to support their claims, focus more on cell (dys)function when the homeostatic adaptation is not working and have made many clarifications to the text. These changes are mostly to the better and the permeability assay is convincing. However, there are a few important inconsistencies and methodological concerns that need to be addressed before this manuscript is ready for publication. Perhaps more importantly, the manuscript still to a large extent contains findings that are not novel although the authors have done an excellent job of making a coherent story of material that has been published previously.

Major points

1. According to Figure 3 the $t_{1/2}$ for DHA incorporation was 4h and to Figure 4 the $t_{1/2}$ for cholesterol occurs after only 1h. At 1h the increase in DHA in the membrane lipids is less than 15% of its peak value. It therefore appears that the cholesterol adaptation precedes the actual lipid remodelling of the membranes, which requires a mechanistic explanation backed up by experimental evidence.
2. The methods the authors use to differentiate between internal membranes and the plasma membrane are problematic. In Figure 5 panel A, the cell areas marked as internal membranes mostly covers the cell nucleus whose envelope is a very loosely packed membrane. Much of the membranes with a lipid composition more similar to the plasma membrane are excluded. If using method 3, as stated in the methods section, for the measurements in Figures 5 the plasma membrane =everything outside the marked area would include a substantial fraction on non-plasma membranes. The actual change in GP-value of the plasma membrane may therefore have been underestimated. Moreover, for many of the cells it appears as if the plasma membrane is not clearly visible and that internal membranes have a higher GP-value than does the plasma membrane so method 1, the line scan, could also fail in reporting on the plasma membrane.
3. Figure S8B. Were the increases of cholesterol in the plasma membrane preparations not statistically significant? Then it should not be stated in the results section that they were increased (line 315). This is important since it is lipid changes in the plasma membrane that would alter cell permeability, not lipid changes in internal membranes. Given the uncertainties of the GP-measurements, it would considerably strengthen the manuscript to have lipidomics data from plasma membrane preparations from cells treated with the DHA+betulin combination.

Minor points

1. Figure 5B & 5H. The colours chosen makes it very difficult to read.
2. Figure 5I. Betulin 500. For consistency it should read +.
3. The term non-apoptotic is confusing. Should it not read necrotic?

4. The caspase inhibitor studies – it is unclear whether the treatment with DHA and betulin was for 72h (line 551) or 24 h (line 552).
5. Active caspase quantification. NEM-treated cells were used as a positive control for thresholding. Please state how since there inevitably were differences between the control cells.
6. Lipid extraction line 597 – is 60 mL correct=
7. Line 698. The word “cell” is missing between “mammalian” and “surface”.

Reviewer #3:

Remarks to the Author:

The revised version of the manuscript by Levental et al., reflects a thorough response to reviewer concerns. Extensive experimental support has been added for each of the major conclusions of the paper, which successfully address the concerns raised and will interest the readers of Nature Communications.

There do remain some very minor comments below:

1. In Authors response 3.4, The authors should add a phrase somewhere in the manuscript that at least acknowledges the slightly increased proliferation in response to low-dose DHA, even if it is not discussed further. Additionally, it was unclear why panel 6C has changed from the previous submission. Is this simply a re-transformation of existing data (as the change in y-axis label implies)? If so, the new data transformation looks like it violates the assumption of homogeneity of variance, and would thus require an alternative statistical test like a robust ANOVA if the authors wish to adopt this new data transformation. Additionally, statistics comparisons between treated/untreated WT CHO cells would be welcome, particularly with a post-hoc test to highlight the specific concentrations at which the differences are significant.
2. In Authors response 3.6, Further modifying the text on page 5 (lines 121 -132) to emphasize the difference in levels of fatty acid saturation between the two diets – level of saturation is only given for the CO diet. As written, the text still highlights the omega 3 vs. omega 6 and chain-length distinction first and foremost.
3. In Authors response 3.9, The authors interpretation of Owen et al. is a valid, but generous, to support the authors current hypothesis. In Owen et al-- While saturated fatty acids are increased up to ~20% after 7 days, this appears to be the peak level of saturated FA reached in the tissue. It is only up 13% at 4 days, and is only up 2% over baseline after 42 days (despite continued increase in PUFA accumulation in that tissue throughout this time course). Thus it may be appropriate for the authors to point out that there is a strong impact of time on lipid remodeling, and if the appropriate time points were not collected then previous studies may have missed this phenomenon.

Dear Dr Mieck,

Thank you and the reviewers for their thorough and positive response to our revisions. We have revised the manuscript to address all the reviewer concerns, including 2 new figure panels and significant revision of the manuscript.

We believe that the reviewers' comments and the associated revisions have significantly improved the manuscript, making it acceptable for publication in *Nature Communications*.

Below please find a point-by-point response to your and reviewer comments.

Editor Comments:

...please address all remaining technical concerns of Rev#2 and Rev#3

We have addressed all reviewer comments, including new experimental data (two main text figure panels)

GP-measurements need to be convincingly done or the requested lipidomic analysis must be provided

This is the new data shown as Figs 5H-I. These observations nicely confirm the previously claimed effect (reduction in plasma membrane packing) with a separate and complementary assay. The reviewer was correct in pointing out that our previous assay was relatively low signal-to-noise and having a more robust assay indeed demonstrated clearer effects. We believe this addition fully addresses the reviewer's comment and strengthens the manuscript.

...the cholesterol adaptation which precedes the actual lipid remodelling should be explained

We have added a mechanistic explanation to the manuscript.

statistics for Fig.6 need to be provided

Done

Reviewer #1

The revision submitted by Levental et al., in my opinion, properly answers the reviewers comments. The manuscript "Lipidomic and biophysical homeostasis of mammalian membranes in response to dietary lipids is essential for cellular fitness" is very solid and timely and it will have a great influence in the field.

We thank the reviewer.

Reviewer #2

In the revised version of their manuscript Levental et al. present more experimental evidence to support their claims, focus more on cell (dys)function when the homeostatic adaptation is not working and have made many clarifications to the text. These changes are mostly to the better and the permeability assay is convincing. However, there are a few important inconsistencies and methodological concerns that need to be addressed before this manuscript is ready for publication.

We thank the reviewer for their supportive comments and constructive criticisms.

Major points

1. According to Figure 3 the t1/2 for DHA incorporation was 4h and to Figure 4 the t1/2 for cholesterol occurs after only 1h. At 1h the increase in DHA in the membrane lipids is less than 15% of its peak value. It therefore appears that the cholesterol adaptation precedes the actual lipid remodelling of the membranes, which requires a mechanistic explanation backed up by experimental evidence.

This is a good point; we were also somewhat surprised by the magnitude and speed of the cholesterol response. One possible explanation is that DHA-containing lipids are highly fluidizing, even at relatively minor relative abundances ¹. Further, these are likely produced first in the ER and may accumulate at the site of synthesis to much higher levels than in total cell membranes. This local perturbation may induce a rapid and potent response by the cholesterol biosynthetic machinery, which has non-transcriptional as well as transcriptional regulation ². It is possible that the early phase of this response is mediated by non-transcriptional regulation of already-available enzymes, whereas the latter phases (including the acyl chain remodeling and increase in saturated lipids) are dependent on transcriptional upregulation of the various machineries.

We have now included this discussion on lines 278-286.

2. The methods the authors use to differentiate between internal membranes and the plasma membrane are problematic. In Figure 5 panel A, the cell areas marked as internal membranes mostly covers the cell nucleus whose envelope is a very loosely packed membrane. Much of the membranes with a lipid composition more similar to the plasma membrane are excluded. If using method 3, as stated in the methods section, for the measurements in Figures 5 the plasma membrane =everything outside the marked area would include a substantial fraction on non-plasma membranes. The actual change in GP-value of the plasma membrane may therefore have been underestimated. Moreover, for many of the cells it appears as if the plasma membrane is not clearly visible and that internal membranes have a higher GP-value than does the plasma membrane so method 1, the line scan, could also fail in reporting on the plasma membrane.

We fully agree and thank the reviewer for prompting us to perform additional experiments that have strengthened the manuscript's findings. To definitively validate the changes in the biophysical properties of the plasma membrane, which were indeed difficult to resolve by C-Laurdan spectral maps, we used fluorescence lifetime imaging microscopy (FLIM) of a probe (Di4) which stains exclusively the plasma membrane (see figures and references in manuscript). Consistent with our previously reported data, we find that only the combination of DHA+betulin resulted in a significantly more fluid plasma membrane.

3. Figure S8B. Were the increases of cholesterol in the plasma membrane preparations not statistically significant? Then it should not be stated in the results section that they were increased (line 315). This is important since it is lipid changes in the plasma membrane that would alter cell permeability, not lipid changes in internal membranes. Given the uncertainties of the GP-measurements, it would considerably strengthen the manuscript to have lipidomics data from plasma membrane preparations from cells treated with the DHA+betulin combination.

The increase in cholesterol induced by PUFA supplementation is not strictly statistically significant (AA-treated cells are $p=0.05$, DHA-treated cells are $p=0.1$ using unpaired t-test); however, given the small sample size, we believe these trends are biologically meaningful. Furthermore, the saturation of the lipids in the GPMVs (PM preps) is significantly affected, thus emphasizing that lipidomic remodeling in response to PUFAs occurs in the PM as well as throughout the whole cell.

We have softened the description in this section.

Minor points

1. Figure 5B & 5H. The colours chosen makes it very difficult to read.

The colors in 5H were changed to increase the contrast and make them easier to read.

2. Figure 5I. Betulin 500. For consistency it should read +.

Thank you, this was fixed.

3. The term non-apoptotic is confusing. Should it not read necrotic?

We have intentionally chosen the term “non-apoptotic” because we believe it is more precise than “necrotic”. This is because we have ruled out apoptosis, but not the several other possible forms of non-apoptotic cell death³. While we agree with the reviewer that necrosis is the most likely alternative, we felt it was best to be conservative in our interpretation.

4. The caspase inhibitor studies – it is unclear whether the treatment with DHA and betulin was for 72h (line 551) or 24 h (line 552).

This has been addressed. The FDA assay was performed after 72hr of treatment to determine the number of viable cells remaining. The cell counting and trypan blue assay was done after 24hr so that dead cells were detected as they detached from the plate rather than being lost with the necessary medium change.

To clarify, we added a line break in the methods between these two assays:

“FDA is a cell viability probe which freely diffuses through cell membranes but is trapped in cells following de-acytation by cytoplasmic esterases in viable cells. The number of viable cells is then directly related to fluorescein fluorescence intensity. For the assay, cells were treated in 96-well plates, gently washed with PBS, and then incubated with fluorescein diacetate (5 $\mu\text{g}/\text{mL}$ in PBS) for 2 minutes at 37°C. The plates were then washed again to remove excess FDA and fluorescence was measured at 488 nm excitation and 520 nm emission. For caspase inhibitor studies, 5 and 30 μM Z-DEVD-FMK were added at the same time as DHA and betulin and treated for 72 h prior to FDA analysis.

To count the number of dead cells, cells were treated for 24 h, and the supernatant harvested. The cells were then trypsinized and combined with the supernatant. 0.4% Trypan Blue was added to an equal volume of cells. The cells were then counted

using a Countess II cell counter (Thermo Fisher), and the number of Trypan-positive cells recorded. For each experiment the number of cells per well were normalized to untreated, unsupplemented cells.”

5. Active caspase quantification. NEM-treated cells were used as a positive control for thresholding. Please state how since there inevitably were differences between the control cells.

NEM treatment was not used for thresholding of caspase positive cells. Rather, it was used as a positive control to validate the assays, with NEM-treated cells being uniformly positive for caspase activation due to wholesale induction of apoptosis.

6. Lipid extraction line 597 – is 60 mL correct=

No, this was a typo. We have now corrected using volume equivalents to be more general.

7. Line 698. The word “cell” is missing between “mammalian” and “surface”.

Thank you. This was fixed.

Reviewer #3 (Remarks to the Author):

The revised version of the manuscript by Levental et al., reflects a thorough response to reviewer concerns. Extensive experimental support has been added for each of the major conclusions of the paper, which successfully address the concerns raised and will interest the readers of Nature Communications.

We thank the reviewer whose comments significantly improved our manuscript.

1. In Authors response 3.4, The authors should add a phrase somewhere in the manuscript that at least acknowledges the slightly increased proliferation in response to low-dose DHA, even if it is not discussed further. Additionally, it was unclear why panel 6C has changed from the previous submission. Is this simply a re-transformation of existing data (as the change in y-axis label implies)? If so, the new data transformation looks like it violates the assumption of homogeneity of variance, and would thus require an alternative statistical test like a robust ANOVA if the authors wish to adopt this new data transformation. Additionally, statistics comparisons between treated/untreated WT CHO cells would be welcome, particularly with a post-hoc test to highlight the specific concentrations at which the differences are significant.

We have since added 2 more independent experiments to this data set. Sidak’s multiple comparison analysis was done on the data, and the only individual data points that were significantly different from each other are SRD12B cells comparing untreated to 80uM or 320uM. None of the DHA concentrations significantly affected the WT CHO cells.

2. In Authors response 3.6, Further modifying the text on page 5 (lines 121 -132) to emphasize the difference in levels of fatty acid saturation between the two diets – level of saturation is only given for the CO diet. As written, the text still highlights the omega 3 vs. omega 6 and chain-length distinction first and foremost.

We have rearranged the wording slightly to reduce the emphasis on the w-3 vs w-6 and focus that section on the differences in abundance of long-chain PUFAs and the general levels of unsaturation between the two diets. We also include the detailed FA compositions of both diets, allowing readers to evaluate the relevant differences.

3. In Authors response 3.9, The authors interpretation of Owen et al. is a valid, but generous, to support the authors current hypothesis. In Owen et al-- While saturated fatty acids are increased up to ~20% after 7 days, this appears to be the peak level of saturated FA reached in the tissue. It is only up 13% at 4 days, and is only up 2% over baseline after 42 days (despite continued increase in PUFA accumulation in that tissue throughout this time course). Thus it may be appropriate for the authors to point out that there is a strong impact of time on lipid remodeling, and if the appropriate time points were not collected then previous studies may have missed this phenomenon.

This is a good point and has now been included in the Discussion.

1 Levental, K. R. *et al.* Polyunsaturated lipids regulate membrane domain stability by tuning membrane order. *Biophys J* **110(8)**, 1800-1810, doi:10.1016/j.bpj.2016.03.012 (2016).

2 Ye, J. & DeBose-Boyd, R. A. Regulation of cholesterol and fatty acid synthesis. *Cold Spring Harbor perspectives in biology* **3**, doi:10.1101/cshperspect.a004754 (2011).

- 3 Tait, S. W., Ichim, G. & Green, D. R. Die another way--non-apoptotic mechanisms of cell death. *J Cell Sci* **127**, 2135-2144, doi:10.1242/jcs.093575 (2014).

Reviewers' Comments:

Reviewer #2:

Remarks to the Author:

The authors have thoroughly addressed the points raised which has resulted in further experiments and clarifications that strengthen the manuscript.

Reviewer #3:

Remarks to the Author:

The authors have more than satisfactorily addressed my concerns